# Smart Lipid-Based Nanosystems for Therapeutic Immune Induction against Cancers: Perspectives and Outlooks

**DOI:** 10.3390/pharmaceutics14010026

**Published:** 2021-12-23

**Authors:** Seth-Frerich Fobian, Ziyun Cheng, Timo L. M. ten Hagen

**Affiliations:** Laboratory Experimental Oncology (LEO), Department of Pathology, Erasmus University Medical Center, 3015 GD Rotterdam, The Netherlands; s.fobian@erasmusmc.nl (S.-F.F.); z.cheng@erasmusmc.nl (Z.C.)

**Keywords:** cancer, immunotherapy, liposomal nanosystems, synthesis, immune reagents, immunoliposomes, cancer vaccine, nano-immunotherapy, clinical trials

## Abstract

Cancer immunotherapy, a promising and widely applied mode of oncotherapy, makes use of immune stimulants and modulators to overcome the immune dysregulation present in cancer, and leverage the host’s immune capacity to eliminate tumors. Although some success has been seen in this field, toxicity and weak immune induction remain challenges. Liposomal nanosystems, previously used as targeting agents, are increasingly functioning as immunotherapeutic vehicles, with potential for delivery of contents, immune induction, and synergistic drug packaging. These systems are tailorable, multifunctional, and smart. Liposomes may deliver various immune reagents including cytokines, specific T-cell receptors, antibody fragments, and immune checkpoint inhibitors, and also present a promising platform upon which personalized medicine approaches can be built, especially with preclinical and clinical potentials of liposomes often being frustrated by inter- and intrapatient variation. In this review, we show the potential of liposomes in cancer immunotherapy, as well as the methods for synthesis and in vivo progression thereof. Both preclinical and clinical studies are included to comprehensively illuminate prospects and challenges for future research and application.

## 1. Introduction

### 1.1. Background

Cancer immunotherapy differs from traditional methods of directly removing a tumor via radiation, surgery, or chemotherapeutic cell kill. In many ways, it is an indirect method, where agents introduced to the body or site of cancer aim at stimulation or harnessing of the immune system for removal of cancer cells, or reversal of characteristic immune dysregulation in cancers [1]. This approach aims to leverage what is available (chemotherapy, radiotherapy, surgery, and other effective treatment modalities) to the scientific and medical community and add to those modalities that can strengthen treatment outcomes.

Cancer immunotherapy lay dormant for many years after its inception in 1892, owing to poor clinical outcomes; hence, tentative adoption in the clinic [2]. However, recent years have seen immunotherapy gaining considerable ground in the clinic, building on increased plausibility and translationality in preclinical investigations, becoming one of the pillars in cancer therapy [3]. These moves necessitate accurate, critical, and contextual review of current literature to encourage further development of this field, pregnant as it is with possibility. Cancer immunotherapy, conceptually, promises lower toxicity, increased quality of life, and importantly, longer-term survival than chemotherapy, with cells gaining immune memory against a type or types of cancerous cells [4]. Immunotherapy has previously consisted of free, unbound agents being introduced to the body systemically or directly into the site of cancerous growth [5], with a view to induce immunogenic cell death (ICD), prevent formation of essential structures to tumor growth [2], or to block negative immune regulation [6]. This poses many challenges, most of which stem from the chemical or biological nature of the active compounds. For this reason, it is desirable to co-administer immunogenic agents in the form of adjuvants, combination therapy, as well as agents which work to target compounds/drugs to the site of disease [7], together lowering systemic toxicity [2].

Nanomaterials have recently seen increased application as drug delivery and targeting agents. This is especially useful in combination therapy, where separate administration of multiple compounds in combination brings into play each compound’s individual pharmacokinetic profile, posing a challenge regarding timing of dosage and inter-individual metabolic differences. Here, nanomaterials add value: packaging agents together can ensure synergistic activity and annul the effects of individual pharmacokinetics of administered agents. This also alleviates the effects of the narrow therapeutic window within which traditional chemotherapeutics operate, by lowering systemic toxicity [8].

### 1.2. Defining “Smart” Nanomaterials

Smart nanomaterials, particularly liposomal nanoparticles, have become popular topics of study within this context in recent years [9]. “Smart” seems to be something of a buzzword in this field, sounding impressive to laypeople, but often having loose relevance or no meaning at all. However, we wish to clarify what is meant by “smart”, and all the applications and considerations that go with that descriptor.

“Smart” can refer to a compound responding to internal (antibody-receptor binding, changes in pH, and increased concentration of reactive oxygen species (ROS)) or external stimuli (change to temperature or exposure to radiation or ultrasound) [10,11,12]. In other cases, “smart” has referred to multifunctionality, tailorability, or an element of control being present regarding release [9]. An extensive summary of smart liposomal therapies is provided in a review by Alavizadeh et al. [12]. We define “smart” as a descriptor of the manipulable and multifunctional drug delivery characteristics of nanomedicines. These characteristics afford researchers the ability to steer, control, or image nanomaterials and their contents or conjugates, from introduction to fate, within a single delivery system. Smart modalities are poised to become a collection of new means of improving treatment outcomes, quality of life, and survival in various cancers, which remain burdensome with persistent high mortality [1].

### 1.3. Immunoliposomes: Semantics and Understanding

When considering the general understanding surrounding the term “immunoliposome”, most publications refer to antibody-conjugated liposomal formulations. This is likely due to the prefix “immuno-”, as in enzyme-linked immunosorbent assay (ELISA), referring to antibodies, and the traditionally-understood immunological functions thereof. We wish to contest this definition, with the specification of immunoliposomes being used only in such cases where the immune system is harnessed in the fight against cancer. Only recently has this been a viable consideration in cancer research [13], with diverse applications.

With the background and justification in this paper, we wish to clarify that the subject of this paper is more correctly stated as “immune-inducing liposomal formulations (IILFs)”.

Therefore, to synthesize a relevant and concise overview of meaningful depth and within a manageable scope, lipid-based nanoparticles and applications thereof in the field of cancer immunotherapy will receive focus in this review. Of interest are the synthesis thereof, conjugation or incorporation of immuno-active elements, pharmacokinetics, chemotherapeutic synergism, and applications in both preclinical and clinical settings. 

## 2. Tumor Immunotherapy

### 2.1. Tumor Immune Response and Escape

The cancer immunosurveillance concept, i.e., the opinion that the immune system could recognize and destroy nascent malignant tumor cells, was first conceptualized by Burnet and Thomas in the twentieth century [14]. However, due to a lack of convincing evidence, this theory was largely ignored for the decades that followed. With better understanding of tumor antigens and the establishment of a completely immunodeficient mouse model, an abundant body of data was generated, which supported the existence of immunosurveillance, rekindling interest in this process. Further research showed that tumors were being modified by the immune environment, which resulted in tumor cells of high immunogenicity being killed by the immune system, but the weakly immunogenic ones being ignored, while simultaneously acquiring other mechanisms for immune evasion. These processes may be due to the inherent genetic instability and rapid adaptability of tumors [15]. New tumor cell variants with reduced immunogenicity are continuously produced and cleared to form a kind of equilibrium. Eventually, the immune system is impaired by the constant stimulation of tumor cells and the balance is upset. It is during this stage that tumor escape occurs and a tumor may progress to clinical relevance. Dunn et al. concluded that three sequential phases exist during interactions between tumors, immune cells, and the tumor microenvironment (TME): elimination, equilibrium, and escape [15,16]. These breakthroughs in the comprehension of tumor immune biology have opened a new chapter in the war against cancer in the new millennium. 

Malignant cells escape from host immune responses via complex mechanisms (Figure 1). Tumor-specific antigens (TSAs) are regarded as vital cornerstones in immunotherapy, consisting of tumor-associated antigens (TAAs), oncogenic viral antigens, and neoantigens, that can sufficiently induce both humoral and cell-mediated immunity. The “ideal cancer antigen” should have highly specific overexpression only in tumor cells, and immunogenicity which could mount a robust immune response. Alas, internal tumor antigens exist, which are usually weakly immunogenic, or develop mechanisms to avoid presentation on the surface of antigen presenting cells (APCs) [17], which fail to activate specific cytotoxic T lymphocytes (CTLs) against tumors. Apart from this, several suppressive factors in the TME can cause immune tolerance and immune deviation, such as: upregulation of immune checkpoints (ICPs) on the tumor cell and immune cell surfaces, low secretion of immune costimulatory molecules, imbalance of the T helper lymphocyte (Th) 1:Th2 ratio, and production of immunosuppressive cytokines. Further to this, immunosuppressive cells can be recruited. These can include regulatory T cells (Tregs), tumor-associated macrophages (TAMs) and myeloid-derived suppressor cells (MDSCs). In addition, chronic inflammation could also stimulate tumor progression [18]. Nutrient competition, including for glucose, amino acids, and fatty acids in the TME can also impact T-cell function and cause T-cell hyporesponsiveness [19,20]. 

### 2.2. Shift of Immunotherapy Strategy

The immune system and cancer progression are both highly intricate and heterogeneous, increasing the complexity in immunotherapy. Early on, scientists paid more attention to enhancing the antitumor immune response in quantity and/or quality. This strategy could be subdivided into two groups. The first approach is passive immunotherapy, which utilizes effector cells and/or molecules of the immune system for direct tumor attack, such as monoclonal antibody and antibody-drug conjugate therapy (ADC), adoptive cell therapy, and genetically engineered T cells (chimeric antigen receptor [CAR]-T, T-cell receptor [TCR]). These passive strategies can boost host immune responses against tumor at a high level. The second approach is active immunotherapy, which can enhance endogenous immune response against tumors. The representatives are antigen/adjuvant vaccines and dendritic cell (DC) vaccines, which can enhance antigen uptake, processing, and presentation to T cells by APCs. This could also extend to cytokines or agents that promote the activity of APCs, such as type I interferons (IFNs), Toll-like receptor (TLR) agonists, and stimulators of interferon genes (STINGs). Although a broad range of applications of the above strategies against various cancers have acquired Food and Drug Administration (FDA) approval, there is only a small proportion of patients, mainly suffering from classical immunogenic tumors, who can benefit from these therapies. Moreover, these immune-enhancing treatments can lead to serious immune-related adverse events (irAEs), such as some off-target toxicities and acute cytokine release syndrome, since the immune system is in supraphysiological level [20,21]. Introduction of these new compounds is accompanied by new side effects which are often mild or moderate, but may be severe and life-threatening, demanding adjustment and development of better therapies or strategies to overcome adverse effects [22,23].

With a deeper understanding of immune escape mechanisms, more explorations emphasize correction of the particular defects or dysfunctions present during tumor progression, and aim to correct or modify blocked antitumor immune responses in comparison with enhancement strategy. Sanmamed and Chen described this shift of immunotherapy strategy as “the age of normalization cancer immunotherapy”. Anti-programmed death receptor (PD) therapy is a typical example of undoing the immune suppression within the TME and has been approved by the FDA for the treatment of various malignant tumors beyond those classically considered immunogenic tumors. Clinical studies show more frequent objective tumor responses than severe irAEs. These exemplify the move within novel cancer immunotherapy strategies, from traditional immune enhancement approaches to immune normalization [21].

## 3. Immune-Inducing Liposomes: Formulation and Characterization

### 3.1. Nanoparticles in Immunotherapy

Nano-immunotherapeutics can have a range of targets and applications due to a large amount of tailorability. Shi and Lammers documented three primary ways to achieve tumor reduction via cancer nano-immunotherapy, over and above direct cytotoxicity [2]. These include cancer cell targeting, immune microenvironment alteration, and peripheral immune system targeting. It is agreed that recent progress made in all three of these areas is indicative of the significant potential held by combinational nano-immunotherapy to improve treatment outcomes in patients [24]. Below, the common applications of nanomaterials in immune therapy are summarized (Figure 2). Excellent reviews exist on the topic [25,26,27], but are beyond the scope of this paper, so only an overview will be provided. We wish to emphasize that the areas of application of liposomes are ideal for cancer immunotherapy: immune induction, synergistic therapy, and drug delivery and targeting [26]. These major strengths are further bolstered by low inherent toxicity [28], earning much clinical interest in recent years [29].

### 3.2. Liposomes

Liposomes are uni- or multilamellar lipid-based vesicular nanoparticles, made up of a phospholipid bilayer encapsulating an internal milieu [30]. Liposomes were first described in 1964 [31], and utilized as drug delivery systems in 1971 [32]. By virtue of the structures of utilized phospholipids, there exist distinct hydrophobic and hydrophilic regions in the core of the liposome and within the lipid bilayer, respectively. Phospholipid bilayers tend to measure approximately 4 nm and liposomes are known by different classifications based on the size and geometry thereof [33]. If multiple lamellae are enclosed in an outer membrane, a liposome is known as a multilamellar vesicle (MLV), whereas single-lamella liposomes, enclosing only a hydrophilic internal milieu, are known as small unilamellar vesicles (SUVs), large unilamellar vesicles (LUVs), or giant unilamellar vesicles (GUVs) due to size; <100 nm, >100 nm, and >1 µm, respectively. This article will address the application of immunoliposomes in the treatment of cancer; therefore, the term liposome, for the purposes of this paper, refers primarily to SUVs. Advantages and disadvantages of liposomes are summarized in Table 1.

#### 3.2.1. Liposome Formulation

Liposome formulation is a topic widely discussed. Most often, research groups will make use of a formulation optimized by themselves for a specific purpose. Common liposome formulations are summarized in Table 2, along with applications thereof. In short, five elements determine the application, functionality and performance of liposomal preparations: (i) natural or synthetic lipid origin, (ii) charge carried by the phospholipid “head”, (iii) acyl chain length, and degree of unsaturation, (iv) the phase transition temperature of the lipids (Tm), and (v) additions to the lipid bilayer or liposome surface [43]. This tailorability is virtually limitless, providing the extreme versatility for which liposomes are well known.

Also included in this are lipidoids; these are lipids modified to suit a certain function [4]. Wang and colleagues developed a library of bioreducible lipid-like materials; essentially, amine-functionalized lipids of many conformations. These were screened for interaction with Cas9:sgRNA complexes with the goal of gene editing in vitro and in vivo. It was resolved that combinatorially designed lipids show lower toxicity and immunogenicity [80].

#### 3.2.2. Liposome Preparation

Three basic steps always comprise the liposome preparation method (Figure 3). These are: (i) combining desired lipids dissolved in an organic solvent and dehydrating, (ii) dispersing the dehydrated lipids in aqueous buffer, and (iii) purifying and homogenizing the product to desired specifications, often using extrusion and/or sonication [8]. In principle, amphiphilic molecules such as phospholipids, when dispensed in water beyond a critical micellar concentration, spontaneously aggregate into structures; namely, bilayers, micelles, or vesicles. The type of aggregation is based on the strengths of present hydrophobic and hydrophilic forces [81]. Synthesis of liposomes developed from an understanding that the lowest-energy state in which phospholipids can exist is as a bilayer. When flattened, the energy state of these bilayers is unfavorable, due to the exposed lipophilic edges. Thus, when rotary vacuum-dehydrated bilayer films are hydrated, the lower bending energy of a vesicular shape causes the formation thereof [81]. During this hydration step, liposomes are formed for the first time, and at this point in the synthesis process, it is crucial for the temperature of the aqueous lipid suspension to be maintained above the phase-transition temperature (T_m_) of the specific lipids in formulation [33]. The Tm is the temperature below which a lipid’s hydrocarbon chains are ordered and extended, and above which, the chains behave fluidly in random conformations [82].

Several reproducible methods have been developed for liposome synthesis. Remaining focused on SUVs, then by a landslide, the most common method is the conventional film-hydration method [7,72,80,83,84,85]. The desired lipids for a formulation are dissolved in an organic solvent, forming reversed-phase micelles. The lipids are combined in this way in desired molar ratios, then evaporated using a temperature controlled, negative pressure rotary evaporator. This forms a thin, uniform lipid film, which is then hydrated with excess aqueous buffer. As described by Bangham, this results in formation of MLVs [31]. The arbitrary-sized MLV suspension is passed through a source of energy, usually in the form of a sonicator and/or extruder. Upon sonication, the inverted micelles entrap the surrounding aqueous solution. Extrusion takes place through membranes of defined size, ensuring uniformity of SUVs at the desired size for the intended purpose [33]. This method forms liposomes which have reasonable stability, can be loaded with both hydrophobic and hydrophilic drugs/compounds, and can be made with lipids functionalized for ligand binding of many kinds. 

The dehydration process is occasionally replaced by freeze-drying (lyophilization), which achieves similar results, with the added benefit of longer preservation times should that be desired [86]. Lyophilization, however, does not replace subsequent steps in the synthesis process, if unilamellar, uniform SUVs are desired, as the lyophilization and rehydration process does not yield uniform particles, in and of itself [87].

Another common way of forming LUVs, GUVs, or MUVs is injection of an organic solution of lipids into an aqueous buffer of choice. This can be achieved by manual or automated injection [88], or by a microfluidic mixing system [89], which is more sophisticated and offers more control over flow rate, conjugation efficiency, and product size. The organic solvent is subsequently removed by dialysis. As with lyophilization, this method is dependent on flow rate and injection needle size to control the size and morphology of the output product. Thus, extrusion and/or sonication remain necessary for uniform, size-controlled SUVs.

An extensive review on newer, more recently developed methods has been published by Has and Sunthar in 2020 [33]. While synthesis methods will not receive the focus here, a summary of three relevant techniques will be provided here [33]. The first is known as the curvature tuning method. This is also based on the spontaneous vesiculation phenomenon, and the curvature theory of lipid bilayers, which, briefly, is responsible for the tendency of a bilayer to preferentially fold into a vesicular shape. This method makes use of a rapid pH change and a lipid mixture containing the zwitterionic lipid lyso-palmitoylphosphatidylcholine, forcing liposome formation in the absence of organic solvents which are otherwise not biocompatible [33,48]. Next, the packed bed-assisted hydration method, whereby in the absence of a post-vesicle formation step (i.e., extrusion and/or sonication), uniform 100 nm liposomes were prepared. This was achieved by passing an organic solvent-dissolved lipid mixture through an asymmetric bed of porous alumina particles. Once dispersed in the alumina bed, the lipid mixture is dehydrated with N_2_ (g), then flushed with an aqueous buffer [81].

The above methods address small-batch generation of SUVs. Electrical methods exist, but function primarily for the generation of GUVs. Large scale methods are also addressed in the above-mentioned review by Has [33]. The following were found to be effective in larger scale preparation of SUVs: the dual asymmetric centrifugation method [90], the membrane contactor method [91], microfluidic hydrodynamic focusing [92], and stationary phase interdiffusion [93], to name but a few. When considering preparation methods, it is important that size and size distribution, membrane fluidity, and surface properties, which have marked effects on biodistribution and clearance [94], are as controllable as possible. It is necessary that reproducibility is achieved between batches for inconsistencies in synthesis and preparation methods to be isolated. These parameters can be assessed with regular post-synthesis measurement of liposome size, surface charge and properties such as drug leakage and/or antibody binding, as relevant [7].

### 3.3. Methods for Liposomal Drug/Ligand Loading/Conjugation

Depending on the desired effect, ligands conjugated onto or loaded within liposomes can be joined in different ways. For the most common ligand types, these methods are outlined below.

#### 3.3.1. Drug Loading

Liposomal drug loading can be achieved, in general, by either passive or active means. Passive drug encapsulation refers to encapsulation during liposome formulation, while active loading occurs post-synthesis [8]. Lipophilic drugs can dissolve and remain within the lipid bilayer, while hydrophilic drugs must be encapsulated along with the internal aqueous milieu of the liposome (Figure 4A). This is often achieved by hydration of liposomes with an agent capable of binding to a drug, causing retention of drug within the liposome, as seen in the popular ammonium sulfate gradient method [95]. Hydrophilic drugs tend to show lower loading efficiency than lipophilic drugs [8]. Active loading, otherwise referred to as remote loading, makes use of a pH gradient between the liposome interior and exterior, trapping ionized drug molecules within and preventing escape from the liposome interior [96]. Drugs conjugated to cholesterol can also be embedded into a liposome membrane. This has been explored using cholesteryl hemisuccinate (CHEMS), a pH-sensitive polymorphic derivative of cholesterol [97].

#### 3.3.2. Peptide Conjugation

Peptides are useful in liposomal formulations due to non-reliance on secondary or tertiary structure for binding to target molecules, or for biological function. As such, peptides can be combined with lipids in an organic solvent and be dehydrated or lyophilized during formation, and still maintain structure and function. Peptide conjugation methods are relatively simple, and most liposomal peptide vaccines, even in clinical trials, are formulated by mixing the elements together preformation, then forming liposomes by normal methods [98].

Traditionally, synthetic antigens were conjugated to liposomes; however, in some cases, low antigen specificity was observed [4]. Thus, ongoing investigations include cancer antigens being isolated from clinical trial patients’ biopsies, and conjugated to liposomes with the aim of a specific immune response being mounted against the endogenous antigen-bearing tumor. 

#### 3.3.3. Cytokine Loading and Conjugation

Cytokines are small (<30 kDa) messenger proteins with positive and negative modulatory effects on many aspects of physiology. These molecules are especially instrumental inducers or regulators of the immune response [99]. For that reason, cytokines have long been molecules of interest in cancer immunotherapy, unfortunately showing unfavorable off-target toxicity in various trials [99]. Thus, these molecules have been loaded into, or conjugated onto liposomes, in hopes of achieving a more targeted effect (Figure 4B). Conjugation can be achieved through coupling of moieties to the phospholipid head groups or, when present, to PEG chain endings [100]. For loading, cytokines such as IFN-γ, interleukin (IL)-12, and IL-2 can be added to the aqueous solution which hydrates the lipid film during synthesis [101,102]. This is a relatively simple method, provided the chosen cytokines are soluble in the aqueous phase for encapsulation by association with anionic lipids [102].

It should be noted that IL-2 can be considered in a similar way to that of antibodies for conjugation to liposome exteriors. IL-2 molecules can be functionalized similarly, or hybridized to contain Ab segments [103,104]. Zhang et al. used a murine fusion protein containing an antibody Fc region coupled to IL-2 head groups, which was later conjugated to a liposome [103]. This was shown to increase circulation time and tumor control via cellular immunity, over that of free IL-2, as well as provide useful methods of liposomal conjugation [105]. The protein molecules were reduced using succinimidyl 4-(p-maleidophenyl butyrate) (SMPB) or dithiothreitol (DTT) for binding to the phospholipid heads external to the liposome, similar to such methods used for antibodies [104].

#### 3.3.4. RNA or Nucleotide Loading

With the success seen lately for liposomal RNA-based drugs, including the first FDA-approved siRNA drug, as well as the recent SARS-CoV-2 vaccination (Pfizer/BioNTech, Moderna), RNA-loaded liposomes have gained attention [30,70]. Packaging of RNA molecules is necessitated by the low stability, high chance of RNase encounters, and negative charge of RNA molecules causing fast clearance [106]. Synthesis involves complexing negatively charged RNA molecules to cationic or ionizable lipids or lipidoid molecules. These can include DOTAP, DSTAP, or lipidoid molecules such as C12-200 [67] or DDA [107]. Cationic variants of cholesterol (DC-Chol), can also be used for embedding in the liposome membrane (Figure 4C), or in some cases, associated with cationic lipids within SLNs (Figure 4D) [68]. Commonly, RNA molecules in compatible buffers are combined with structural and RNA-complexing lipids using a microfluidics system to collide the solutions under controlled conditions such as flow rate, pH, and mixing chamber size. These systems are preferred in the case of RNA-encapsulated liposomes, due to higher encapsulation efficiency. Importantly, the instability of RNA molecules places limitations on the choice of complexing method, in terms of solvents and buffers. RNA molecules can also be complexed to liposomes externally, with addition of a functionalized RNA solution to preformed liposomes. Blakney et al. carried out a detailed comparison between these two methods, observing that interior encapsulation of RNA with C12-200 and DOTAP lipids was protective from RNAse, but not with DDA, which showed quite the opposite phenomenon. These results emphasize the importance of lipid choice in formulation for a desired purpose [70]. 

### 3.4. Antibody Conjugates

#### 3.4.1. Antibody Structure

Understanding liposome loading and conjugation techniques begins with an understanding of the conjugates’ structures. For example, in the case of immunoliposomes, it is crucial that liposome conjugation does not hinder antigen binding. For proteins, where primary, secondary, and tertiary structures may have functional relevance, it is critical to ensure correct orientation, as well as to consider the use of appropriate buffers which (i) contain ideal tonicity for retention of tertiary structure and (ii) are of appropriate pH, preventing unwanted ionization or reduction of residues, potentially creating additional and unwanted liposome–Ab interaction sites.

Antibodies are the most common versatile of liposome conjugates and use thereof in the nano-immunotherapy field has increased dramatically in recent years, with considerable success [108,109]. This success began with the therapeutic use of free monoclonal antibodies. Antibody-based anticancer therapies have undergone great development following the advancement of monoclonal antibody (mAb) manufacturing techniques and the advent of genetic modification of murine mAbs to form chimeric mouse-human mAbs [110]. This technology has advanced parallel to the development of liposome synthesis, together contributing to the considerable progress in the field of immunoliposomes in recent years.

Antibodies, i.e., immunoglobulins, are soluble, globular protein molecules capable of binding antigens with varying specificity, for opsonization and immunogenic elimination of foreign bodies. Antibodies are made up of two light (_L_) and two heavy (_H_) chains, each _L_ chain made up of one variable (V) region, and one conserved (C) region, and each _H_ chain made up of three C regions and one V region. The V_L_ and C_L_ chains, as well as the V_H_ and a single C_H_ region of the _H_ chain, together, make up the Fab region. This is situated at the top of the “Y” shape of the antibody, and forms the functional, or antigen-binding site. The Fc region, the stem of the “Y” shape, is made up only of _H_ chain fragments, and is the site of the antibody which, when circulating, attracts immune cells for binding and elimination thereof, as well as activation of complement pathways. The end of the binding site, made up of a V_L_ and a V_H_ chain, is known as the single-chain variable fragment (scFv) [65]. Lastly, the VH chain alone is known as a single-domain antibody (sdAb). This is the smallest functional unit of an antibody, also known as a nanobody [75].

Primarily, the IgG class of antibodies is used therapeutically for its relative abundance in the blood, lymph, peritoneal and cerebrospinal fluid. IgGs can be further subdivided into IgG1–4, IgG1 and 3 generally targeting protein antigens, and IgG2 and 4 generally targeting polysaccharides. For use with liposomes, whole antibodies often need to be modified or separated into chains or regions. This can be to reduce steric hindrance on the liposome surface, to increase stealth characteristics of the liposome, or to prevent Fc-mediated recognition by the immune system. Constant domains of antibodies are not directly involved in antigen recognition and binding, but Fc domains may be recognized by the mononuclear phagocytic system (MPS) in circulation, resulting in faster clearance. This effect was demonstrated early in immunoliposome development by Iden and Allen, where anti-CD19 mAb-conjugated immunoliposomes showed negligible targeting and greatly increased clearance compared to untargeted, PEGylated liposomes [111]. Modifications are commonly carried out by enzymatic digestion at particular residues using pepsin or papain, or chemical reduction of disulfide bonds, using reducing agents such as 2-mercaptoethylamine (MEA), DTT, tris(2-carboxy)phosphine (TCEP) and dithiobutylamine (DTBA) [112]. Cleavage sites and antibody structures are shown in Figure 5A. 

As an alternative, advancements in protein engineering and expression have enabled the generation of novel classes of recombinant human antibody fragments. For example, scFv and sdAb. scFv fragments are not obtainable via cleavage, but rather, via a phage display using bacterial cultures to generate hybridoma cultures capable of generating proteins of interest [65]. These have been used with varying success in targeted liposomal therapy [65,80,113,114,115]. Advanced technologies also make it possible to produce diabodies, which comprise two different scFvs simultaneously targeting two distinct antigens. The dual-targeting capacity gives diabodies higher avidity. Rabenhold et al. showed that diabodies of scFvs targeting CD105 and fibroblast activation protein (FAP) may be superior to scFv in targeting ligands of drug carriers, without influencing toxicity [116]. Lastly, sdAbs, or nanobodies, show certain possible uses in this area. Obtainable via phage display, or from camelids or transfected mice, nanobodies’ smaller size assists epitope recognition, extravasation, stability, and reduces immunogenicity [117]. However, this small size can also hinder the efficacy of nanobodies, as molecules under 50 kDa are rapidly cleared from the bloodstream by the kidneys, and ADCC, as well as Fc-mediated complement-dependent cytotoxicity are lost. Depending on needs, this may reduce therapeutic capacity of the nanobodies, which supports conjugation of nanobodies to liposomes for delivery [117,118,119].

#### 3.4.2. Antibody Modifications Allowing for Binding

Antibody preparation techniques involve functionalization with a ligand that allows covalent or non-covalent association with the lipid molecules forming the bilayer [94,120]. Basic subdivisions include utilizing reduced, existing sulfhydryl groups on cysteine residues, or ligating a sulfhydryl group to a primary amine, found on lysine residues. These two methods have been achieved in multiple ways, but the underlying chemistry is similar. Figure 5B shows a simplified summary thereof, explained in greater detail in reviews by Manjappa et al. [120] and Di et al. [94]. Typical direct linkages include the use of 2-iminothiolane (Traut’s reagent), N-succinimidyl 3-(2-pyridyldithio)propionate (SPDP), or N-succinimidyl-S-acetylthioacetate (SATA) for mAb functionalization, or DTT, mercaptoethylamine (MEA), or dithiobutylamine (DTBA) for reduction of available cysteine residues on Fab’ or F(ab)_2_ fragments [94,121,122]. Indirect linkages can also be made using a molecular linker, for example, p-nitrophenol (p-NP), cyanuric chloride, biotin-avidin, or folate-PEG-cholesteryl hemisuccinate (CHEMS) [94,100,123]. Conjugation of whole antibody via selective functionalization of the Fc region is also possible. This ensures correct and preferred orientation of the antibody with antigen-binding sites facing outward, away from the liposome [124]. This improves upon early conjugation methods and addresses a common concern amongst immunoliposomes, that incorrect mAb orientation could impede binding, increase immune recognition, and accelerate clearance from circulation [7,125]. 

#### 3.4.3. Coupling Techniques

Many conjugation methods have been tested in the past, to link antibodies to liposomes, and in articles falling within the scope of relevance of this review, two methods were in the majority—conventional coupling [72,83] and post-insertion [84,85,126,127,128], with the latter taking precedence in recent years. These methods will be distinguished and explained alongside the benefits and drawbacks thereof.

Conventional coupling relies on the formation of a liposome with linking molecules incorporated into the primary structure, at the commencement of synthesis. For example, polyethylene glycol (PEG) chains with maleimide linkers, anchored to phospholipids (Mal-PEG lipids) are included in the original liposome formulation, and a modified (reduced, linked, extended) antibody, Fab, scFv, or sdAb molecule is added afterward via covalent binding to maleimide groups [126]. Because conventional coupling involves inclusion of the functionalized PEG molecules into the primary formulation of the liposomes, this method prepares liposomes with maleimide molecules extending both outward and inward relative to the lipid bilayer, potentially squandering antibody binding sites and diminishing the antibody-maleimide conjugation capacity of the liposomes. Post-insertion involves the formulation of liposomes lacking linker molecules. Whole mAb or fragments thereof are formulated as DSPE-PEG(2000)-Mal micelles and added to preformed liposomes. This ensures availability of all maleimide linkers on the outside of the liposome, maximizing conjugation capacity and efficiency. Nanobodies have also been coupled in this way [118,119].

Post-insertion is regarded as the newer, more advanced, superior method, which allows more control [7]. Antibodies are also limited due to reliance on tertiary structure for activity, which needs to be maintained throughout the synthesis and conjugation process. This means that avoidance of organic solvents and incompatible buffers is of high importance when preparing immunoliposomes [84,85]. In our experience and as documented [7], post-insertion yields a higher efficiency and does not affect stability or physicochemical properties of the liposomes. Additionally, in the case of loading drug and targeting with antibodies simultaneously, it is known that post-insertion is superior for use in manufacturing immunoliposomes, when considering flexibility, control, time, and relative ease [111], as the environment of the antibodies or fragments thereof can be controlled up until addition of the fragments to liposomes [126].

## 4. In Vivo Progression of Liposomal Nanoparticles

### 4.1. Uptake, Clearance, and Biodistribution

Being a drug delivery method, an understanding of the movement of liposomes through the body from administration until exit is extremely important. Liposome factors which play a role include size, surface charge, membrane fluidity, and external conjugates [94]. Larger (500–5000 nm) MLVs are recognized more easily than smaller (<100 nm) SUVs, while charged particles are cleared faster, unlike neutral particles [129]. Membrane fluidity can affect the stability of the liposomes, and membrane conjugates can cause retention for longer periods upon antibody or ligand binding [7,44]. 

Upon entrance to the bloodstream, complement proteins and opsonins bind to the liposomes [129]. Opsonins attract phagocytic cells for degradation, which recruit complement proteins to form a membrane attack complex, which destroys liposome membranes and causes entrapped compounds to leak out [129]. This leads to degradation and clearance by the reticuloendothelial system (RES), as well as complete prevention of biological activity or targeting, or even unwanted toxicity [130]. Some attempts were made to overcome the above by saturating the biological processes underpinning these effects. This was done by administering blank liposomes prior to liposomes with therapeutic value, to allow the therapeutic liposomes to take desired action unhindered by innate immune processes [129,131]. Immune evasion is better achieved with the addition of immunologically inert PEG chains on the exterior of the liposome, known to impart “stealth” characteristics to the liposome [94], by introducing serum protein-inhibiting steric hindrance on liposome surfaces [129]. This approach has been extremely popular in reducing toxicity and antigenicity, and was instrumental in securing the FDA approval of Onivyde^®^ (PEGylated irinotecan/fluorouracil liposomes; Merrimack Pharmaceuticals, Cambridge, MA, USA), Lipo-Dox^®^, Doxil^®^ (PEGylated doxorubicin (DOX) liposomes; Sun Pharmaceutical Industries, Mumbai, India and Baxter Healthcare Corporation, Deerfield, IL, USA, respectively), and others [43]. Combination of liposomal irinotecan with 5FU and folinic acid improved overall survival in gemcitabine-pretreated pancreatic cancer patients, demonstrating the potential efficacy of liposomal chemotherapy [132]. The stealth characteristic ensures longer circulation time and as a result, increased tumor-site accumulation [129]. PEG molecules can vary in length [7], and this has effects such as neutralization of charge with longer PEG chains (PEG(350)-PEG(2000)). This is likely due to the masking of exposed phosphate groups on the lipid head responsible for the negative zeta-potential in many cases [133]. Kowalska et al. show that small modifications such as PEG chain length can have downstream effects on uptake, and thereby, efficacy and toxicity [133].

Unfortunately, PEGylation does not avoid all immune detection: an anti-PEG IgM response was noticed after repeated administration of PEGylated particles, resulting in accelerated blood clearance [94]. This is particularly restrictive for immune-inducing liposomal therapies, sometimes requiring multiple administrations to achieve an antitumor response [94,129]. Further to this, it is common for adsorbed proteins on the surface of nanomaterials, including liposomes, to form a protein corona. This can mask and alter surface properties and binding capacity in the case of ligand-conjugated liposomes. This necessitates thorough characterization of liposomes in all relevant media [130].

Various studies have assessed biodistribution of liposomes, many showing contrasting results [129]. Logically, this depends strongly on the type of liposome in question, the efficacy of the targeting method, the degree to which the enhanced permeation and retention effect (EPR) comes into play for a particular cancer [9], and importantly, the route of administration [134]. Briefly, within 72 h post-administration intravenously, the spleen, kidneys, and liver were the most common areas of localization [129]. However, other routes including intraperitoneal, intramuscular or subcutaneous administration can result in retention of up to 80% of liposomes at the site of administration. Liposomes remaining within the site of administration are readily cleared by the MPS and therefore, poorly bioavailable [134]. We believe it would be most valuable to, as a matter of standard practice, characterize this parameter for each new liposome formulation, such that on a case-to-case basis, accumulation can be recorded and reasons for what is observed can be sought. For example, if an immune depot-forming liposomal formulation is used, it may even be desirable for it to remain near the site of administration, especially if given intratumorally [2].

#### 4.1.1. Peptide-Conjugated

Peptide conjugation involves mimicry by liposomes, of antigen presenting cells. The objective is for B-lymphocytes to manufacture specific antibodies against these peptides, or for the peptides to be delivered intracellularly to APCs, be broken down and processed for presentation in the context of MHC I and II, and cell-based immunity to occur [134]. The extent to which externally-conjugated peptides are recognized by cells of the immune system will greatly affect the clearance rate of such liposomes, whereas peptide-loaded liposomes will follow similar clearance patterns to regular or PEGylated liposomes [134].

#### 4.1.2. Nucleotide-Conjugated

Nucleotides have previously been added to formulations as adjuvants for induction of immunity and stimulation of innate immunity pathways via binding to TLR3, 7, and 8 [69]. These molecules are also known as pathogen-associated molecular patterns (PAMPs), which are macromolecules expressed commonly by a range of pathogens. Recognition of single-stranded, double-stranded, and small interfering ribonucleic acids (ssRNA, dsRNA, and siRNA, respectively) takes place in the endosomes of specialized phagocytic cells known as plasmacytoid DCs (pDCs) [135]. Upon recognition, IFN-α and -β, as well as pro-inflammatory cyto- and chemokines are released, including TNF-α, IL-6, and IL-12 [69]. In some cases, these effects may be desired as adjuvant systems. This is also seen in the use of CpG oligonucleotides, an immunostimulatory unmethylated dinucleotide recognized by TLR9, causing DC maturation, antigen expression, and increased cytotoxic immune response [136]. Thus, in the case of RNA-based vaccination, it is necessary for the RNA molecule to stay intact for successful translation. RNA must be protected from contact with undesired compartments of the body until arrival at site of disease [70].

#### 4.1.3. Antibody-Conjugated

What were traditionally known as immunoliposomes have been extensively studied in terms of clearance and circulation. This is because antibodies are immunogenic by nature, and the therapeutic use thereof requires structural tweaks overcoming this immunogenicity, both to the antibodies and the liposomes. As discussed, this can involve removal of the Fc fragment and addition of PEG to the outside surface of the liposomes.

Because antibodies primarily function as targeting moieties in this case, it is essential that a balance between effective targeting and low clearance is found. This entails finding the optimal antibody conjugation density that achieves desired targeting effects, without raising clearance rates of immunoliposomes. Certain antibody fragments, such as scFv, may be less immunogenic, thereby extending half-life. Smaller ligands also allow for higher loading capacities per liposome, and effective orientation of targeting ligands, likely leading to overall improvements in efficacy [65]. Depending on antibody type and density of antibody loading, plasma half-life in vivo is typically shorter for non-PEGylated immunoliposomes than PEG liposomes. Furthermore, in an investigation by Cheng et al., in which mAbs, Fab fragments, and scFv fragments complementary to the same epitope were compared, scFvs showed the highest clearance because of His- or c-myc-tags from synthesis attracting immunity and causing higher liver uptake. There was negligible difference in clearance observed between the mAb and Fab fragment-loaded liposomes [137]. Other studies have shown that Fab fragments have superior therapeutic responses, achieving longer circulation times than mAbs [125]. In contrast, there is other evidence suggesting that scFv-immunoliposomes show the same clearance pattern as liposomes with no antibody [138]. Rodallec et al., in a study comparing biodistribution of immunoliposomes to liposomes, showed that a slightly higher concentration of liposomes than immunoliposomes was found in tumor tissue over time in some tumor types, though no significant increase was noted [139]. This was strongly dependent on the variation between tumors in question, as different tumors resulted in a 10-fold increase in localization, but still no significant difference between immunoliposomes and liposomes. Aspects of tumor variation can include, among others, changes in vascular density, permeability, size, location, and stage of progression [140]. Once again, drug–antibody combination-loaded immunoliposomes showed higher efficacy than the non-antibody-labelled counterparts [141]. This was not due to higher accumulation at the tumor site; more likely, it is due to the therapeutic (not targeting), capacity of the mAb, in this case, trastuzumab.

Immunoliposomes of different Ab concentrations were compared in terms of cytotoxicity and cell uptake, and it was seen that of three tested concentrations, the highest was at best the same, due to binding site blockade effects and steric hindrance [83]. The same effect was observed by Li et al. [142]. In our opinion, the above listed studies prove one thing: the lack of standardization across antibody handling methods and liposome formulation make it difficult, if not impossible, to compare inter-study results. In any study pertaining to liposomes, characterization should always be carried out between batches, and formulation techniques should be optimized on a case-by-case basis.

### 4.2. Passive Targeting via the EPR Effect

#### 4.2.1. EPR-Mediated IILF Targeting: Necessary, but Insufficient

The EPR effect describes a phenomenon whereby particles can extravasate from the bloodstream towards a tumor due to leaky and compromised peritumoral vascular endothelial cells. This, combined with lymphatic insufficiency within a tumor potentiates passive retention of particles localized in this way. Tumors are unique, heterogeneous, and are more permeable at different phases of progression, as well as at different sites within the body. Thus, the EPR effect is unreliable when considering the use of immunoliposomes in widespread cancer therapy [9,141]. Moreover, reliance only on the EPR effect for efficacy in nanomedicine has yielded little clinical success [143]. Therefore, the EPR effect is both necessary and insufficient for therapeutic targeting on its own, and an optimal balance is necessary between the conjugation of active targeting agents and the maintenance of stealth characteristics, like sufficient PEG coating; in other words, between targeting effects and avoidance of clearance by the RES.

Kirpotin et al. compared targeted and untargeted stealth liposomes in terms of uptake, tissue penetration, and accumulation. Increased therapeutic efficacy was observed with targeted liposomes, though not due to greater accumulation at disease site. Non-targeted liposomes localized to the tumor stroma, extracellular space and even to tumor-associated macrophages, while targeted immunoliposomes localized to the cytoplasm of cells bearing the corresponding antigen after being internalized. They suggested that the increased therapeutic efficacy of targeted liposomes is rather due to release of contents upon uptake by tumor cells, or greater penetration into deeper layers of a tumor [141]. However, efficacy and penetration depend strongly on the nature of particular cell or tissue being targeted.

Traditional chemotherapeutic paradigms may suggest tumor accumulation to be a function of dosage size, but for nanomaterials, tumor site accumulation is dependent on circulation time, particle size and shape, as well as dose administered [141,144]. Investigations have revealed that accumulation kinetics were indeed affected by plasma concentration, but extent of accumulation was not. Additionally, binding site saturation was ruled out as a possible cause for the plateau because plasma concentrations correlated well with liposome accumulation [141]. Accumulation depended more on dynamic tumor morphology and permeability than liposome characteristics.

An important reason for the non-reliability of EPR as a targeting mechanism is the discrepancy between murine models (where successful passive targeting was seen) and humans [143]. Particular aspects are also commonly heterogeneous within and between humans, making the murine model insufficiently translatable. These include heterogeneity in vascular endothelial fenestrations, irregular blood supply to parts of tumors causing insufficient nutrient levels within the tumor and/or waste product build-up, heterogeneous basement membrane and pericyte presence, and importantly, irregular extracellular matrix (ECM) condition and density. Of the many effects caused by the above, interstitial fluid pressure (IFP) is most notable in the context of the EPR effect [143], as this shows the most direct effect on the extent of EPR, while simultaneously being affected by multiple factors. Bertrand et al. describe the many driving forces of the EPR effect mathematically, taking into account vessel architecture, interstitial fluid and ECM compositions, phagocyte infiltration, and presence of necrotic domains. They also account for nanoparticle (NP) properties such as size, shape, charge, and surface characteristics. It became clear that the expectation for a single, uniform phenomenon to have the same effect between different individuals and different particles, when governed by so many factors, was not reasonable. Acknowledging this, Bertrand and colleagues include “Black box” variables in their equations, which include unpredictable tissue conditions and NP effects, as well as inter-individual variation [145].

#### 4.2.2. Therapeutic Targeting of the EPR Effect

In this light, it would be beneficial to develop a method of clinically predicting or normalizing the state of tumor vasculature before drug administration. In other words, characterization of the presence and extent of EPR. This would give rise to a new diagnostic method, as the success of nanomedicine is inextricably linked to the effects of EPR. Alternative therapeutic methods linked to the EPR effect have been proposed, such as the remodeling of the peritumoral environment, including the vasculature and the ECM. This was done with some success using losartan, which inhibits collagen I production, permeablizing the TME [146], but was also achieved with photo-immunotherapy [147]. Both of these increased the efficacy of liposomal anticancer formulations [145]. 

Haider et al. describe the use of sildenafil in a similar manner, leveraging its activity as a vasodilator and smooth muscle relaxant for increased tumor targeting of drugs [148,149]. These effects extend to all molecules, not only nanomedicines; for example, free DOX administered with sildenafil showed increased accumulation and therapeutic efficacy [150]. Naturally, it is important to consider the limiting systemic effects of a drug such as sildenafil: uses of such drug combinations will be limited to patients not sensitive to increased vasodilation [148]. 

Islam et al. investigated a range of agents which could potentially enhance the EPR effect with the goal of potentiating delivery of nanomedicines to disease sites [151]. They tested sildenafil, but also isosorbide dinitrate (ISDN) and L-arginine. These agents target various aspects of nitric oxide-mediated vasodilation-inducing pathways described in detail by Maeda [152]. Briefly, they showed an increase in tumor delivery and efficacy of polymer-based nanomedicines against an in vivo advanced colorectal cancer model, of up to 4-, 2-, and 2-fold (from treatment with ISDN, sildenafil, and L-arginine, respectively). 

Other efforts have targeted normalization of tumor vasculature and improvement of transvascular convective movement of nanomaterials once internalized [145]. Considering the above, it is clear that we have emerged from the era of reliance on EPR for efficacy, and we currently look towards EPR as a cotherapeutic target in cancer nanotherapy.

## 5. Therapeutic Strategies of Antibody-Conjugated Liposomes

As mentioned, due to the unique and favorable pharmacokinetic profiles of liposomes, together with the rapid development of mAb-based cancer therapies, oncologists see immunoliposomes as a promising strategy, which has typically been utilized to directly target vital signaling pathways in tumor progression, or to co-deliver antitumor drugs or modulating agents, but may in future, show potential in more diverse therapeutic areas (Figure 6).

### 5.1. Direct Target Transmembrane Signaling on Tumor Cells

Immunoliposomes may be used to target surface antigens on malignant cells, to induce apoptosis by direct transmembrane signaling without immune effector mechanisms [153], which results from mAbs disturbing the interaction between an effective ligand and the receptor. Abnormal expression of the molecules by tumor cells and limited soluble targets in circulation both promote the feasibility of immunoliposomes. Crosslinking between the Fc region of IgG molecules on immunoliposomes to Fc receptors (FcRs) expressed by a variety of cells in the TME can help to generate a robust intracellular signal-inducing death of malignant cells [110]. A range of specific liposomal antibodies exist which aim to inhibit key regulators in tumorigenesis. For example, membrane-bound hepatocellular receptor tyrosine kinase class A2 (EphA2) is overexpressed in various types of cancers and related with several signaling pathways of invasion and metastasis [154]. Merrimack Pharmaceuticals established a taxane-liposome coated with anti-EphA2 scFvs, which showed improved efficiency and specificity [155].

### 5.2. Interaction with the TME

The TME has long been considered a target for cancer treatment, as non-tumoral cells in the TME are likely to be more genetically stable; thus, less prone to developing drug resistance [24]. Early interference with the TME may be particularly beneficial in the treatment of malignancies that are likely to metastasize. TME-targeted immunoliposomes work by modification of defective cells in the TME, such as new intratumoural blood vessels and stromal cells, rather than directly killing tumor cells. Tumors can be controlled to an extent by inhibition of angiogenesis or by inhibition of the activity of tumor-associated fibroblasts to reduce the production of matrix components. mAb-conjugated liposomes that bind to overexpressed receptors on tumor cells or other cells in the TME are common strategies to achieve this. For example, Jain et al. coated docetaxel-loaded liposomes with ligands that bind specifically to vascular endothelial growth factor 2 receptors (VEGFR-2), a key angiogenic factor which is commonly upregulated in new peritumoral blood vessels, in breast cancer therapy [156]. This prevents synthesis of new blood vessels, depriving tumors of essential nutrients. Improved antitumor efficacy was achieved when used in combination with docetaxel, likely caused by an indirect effect from anti-angiogenic activity of these mAbs.

### 5.3. Antibody-Drug Conjugated Liposome

Antibody-conjugated liposomes, for the remarkable ability thereof to deliver various moieties selectively to tumor cells, fall under the description of Paul Ehrlich’s “Magic Bullet”, especially when using these systems to deliver cytotoxic drugs, which have been studied for decades [157]. Co-delivery of tumor-specific monoclonal mAbs with highly cytotoxic payloads in liposomes can theoretically improve the efficacy/toxicity balance of antibody drug conjugates (ADCs). Immunoliposomes could provide both direct targeting of tumor cells as well as “bystander killing”, whereby released drugs diffuse to adjoining tumor cells [158]. Tumors expressing antigens expressed at lower concentrations could also be recognized and killed by co-delivery of a highly cytotoxic drug. Additionally, there is no limitation of drug/antibody ratio when using liposomal delivery (whereas there is typically a limit of less than 10 molecules per antibody for ADC, or antibodies may be inactivated). Furthermore, indirect linkage of drugs to mAbs could alleviate difficulties with stability and immunogenicity of immunoconjugates [159]. 

Importantly, tumor progression is extremely complex and mediated by multiple pathways. Even in the face of breakthroughs in targeted therapies, resistance to specific pathway inhibitors can develop, with the emergence of cells that activate alternative or compensatory signaling pathways. In reality, it is not advisable to use a singular targeted therapy, due in part to the variations of TAAs among individuals [17]. The majority of research into immunoliposomes in recent years has been in combination with cytotoxic drugs proving reluctance to shift away from traditional chemotherapeutic modalities [100,159]. This multifunctional liposomal ADC could overcome limitations of both nonspecific cytotoxic drugs and specific, but often ineffective, mAbs, which have been recognized as hopeful strategies.

Plasmid DNA and siRNA have also attracted recent attention in cancer immunotherapy, with several antibody-nucleic acid conjugates currently in preclinical and clinical trials. For example, Yingying et al. designed an anti-endoglin liposome containing an α,3-galactosyltransferase plasmid which showed efficacy in several cell lines, including human bronchial carcinoma, human colorectal carcinoma, and human hepatocellular carcinoma cell lines [159].

## 6. The Novel Concept of Immune-Inducing Liposomal Formulations

Although great progress has been made in the area of immunoliposomes, ambitions for liposomal immunotherapy are greater than this. With increasing insights into host antitumor responses, researchers have gradually shifted attention from “utilizing immune techniques” to “modulating the immune response”. Novel IILFs are blooming, with focus on tumor cell-intrinsic properties, including absence of antigenic proteins, antigen presentation, and T-cell exclusion from signaling pathways, as well as tumor cell-extrinsic factors, consisting of the other cells, molecules and nutrients present in the TME (Figure 6). 

### 6.1. Cancer Vaccines

Cancer vaccines are made up of antigenic epitopes derived from TSAs, which aim to deliver these to APCs for presentation in the context of MHC-molecules, to CD4^+^ and CD8^+^ T-lymphocytes. This induces the immune system, particularly CD8^+^ cytotoxic T lymphocytes [160]. Liposomes are popular immuno-adjuvants and delivery systems, and different types of liposomal cancer vaccines which have emerged in recent years are discussed below and presented in Table 3. 

#### 6.1.1. DNA and RNA Vaccines

Nucleic acids encoding proteins or peptides which are defective and/or lacking are delivered to the cytoplasms (RNA) or nuclei (DNA) of targeted cells, with the aim of inhibition of key survival pathways of tumor cells. RNA vaccines are capable of bypassing MHC class restriction, eliciting immunogenicity without the need for adjuvants. This occurs via its immunogenic potential as a toll-like receptor agonist. These RNA vaccines are relatively easy to produce and store. DNA vaccines are another attractive approach due to stability, safety, and simplicity of production. Additionally, it was reported that the probability of DNA vaccine integrating into the genome of patient cells is as low as spontaneous mutation [161]. 

It is acknowledged that naked DNA/RNA molecules can be taken up by many types of APCs, including myocytes, monocytes and DCs, but are easily degraded by plasma or tissue enzymes, especially RNAses. Further, the transfection efficiency is relatively low [162]. Therefore, efforts are focused on developing stable and targeted delivery systems which could protect nucleic acids from premature degradation, promote site-specific release, increase uptake in tumors, and reduce systemic side-effects. In this way, non-viral vector-cationic lipid complexes have recently become strong competitors [163], due to electrostatic interactions and self-assembly [164]. 

For instance, Mai et al. established a cationic liposome composed of DOTAP/Chol/DSPE-PEG. DOTAP can come into direct contact with the cell surface through electrostatic interaction, is easily taken up by DCs, and helps DCs to present antigens. They used protamine with positively charged polycations to condense mRNA-encoded cytokeratin (mCK19), a specific protein prevalent in lung cancer cells, for complete encapsulation within liposomal carriers. The results show that this liposomal mCK19 vaccine could effectively promote antigen uptake, DC maturation, cytokine secretion and antitumor effects in vitro and in vivo [165].

#### 6.1.2. Peptide Vaccines

Liposomal peptide vaccines, defined as highly purified tumor-associated peptides, possess strong safety profiles, ease of production, and low production costs, relative to classical live-attenuated or inactivated vaccines. However, difficulties include poor immunogenicity, rapid degradation, relevant immune tolerance, and low bioavailability in vivo. Short peptides (<15 amino acids) can bind to MHC class I molecules directly, without APCs, which may not provide sufficient co-stimulation, leading to tolerogenic signaling and T-cell anergy. Because of these, the specific peptide sequences can be elongated to generate multivalent synthetic long peptides (SLPs), containing both MHC class I and II epitopes. These can elicit a balanced induction of both CD8^+^ and CD4^+^ T cells [17]. Efforts to improve the potency and quality of peptide vaccines have also included synthesizing constructs with other immune adjuvants and delivery systems, like liposomal nano-platforms. 

For example, Naghibi et al. designed a novel liposome with two multiepitope peptides from human epidermal growth factor receptor 2 (HER2/neu (P5 and P435)). DOPE was used to enable the fusion of liposomes, and presentation of antigens via phase transition induced in the acidic pH of endosomal compartments. Stronger immune responses with CTLs and tumor regression were detected in vivo [166,167]. 

#### 6.1.3. Cooperative Combat against Cancer 

Taking the precision and dynamic nature of the immune system into account, multitargeting strategies should be considered. Co-administration of MHC I- and II-restricted tumor antigens and adjuvants like TLR and nod-like receptor (NLR) agonists, in one liposome delivery system, can effectively elicit more efficacious responses in comparison with antigen presentation alone. For instance, Jacoberger-Foissac et al. designed a liposome platform consisting of an MHC class I epitope peptide from human papillomavirus (HPV) 17 E7, MHC class II epitope peptide, which is universal and derived from influenza virus, and an immune agonist. This could induce potent Th1-oriented antitumor immunity and the migration of skin DCs to draining lymph nodes [76,168]. Another example of combination therapy is to combine TSAs and specific mAbs targeting APCs. For example, LAG-3 is one of the main adaptive immunity ligands that binds to 15–20% of MHC II molecules on the surface of DCs, followed by the priming of CD4^+^ and CD8^+^ T cell-mediated antitumor responses. Mohammadian et al. produced a LAG3-Ig/P5 peptide liposome which had interactions with MHC II molecules on the surface of DCs, which was able to induce the activation and maturation of DCs. These novel IILFs enhanced CTL-mediated cytotoxicity, elicited higher percentage of CD8^+^/CD4^+^ T cells in the tumor area, and increased cytokine secretion (e.g., IFN-γ) [127].

### 6.2. Reversal of Immune Suppression in the TME

Immune resistance within the TME exists due to dominant suppression through immune inhibition mechanisms. Inhibitory receptors are commonly overexpressed on the surface of tumor cells because of inflammation and antigenic stimulation [85,182]. Blocking inhibitory receptors can at least partially reverse T-cell exhaustion and improve antitumor T-cell responses. There are several immune checkpoint inhibitors that have been approved by the FDA, including blockers of the T-cell immune checkpoint receptor PD-1 or its ligand PD-L1 which are commonly overexpressed in tumor cells, as well as CTLA-4, an immune checkpoint receptor on T cells, thereby adding a new tool to the box in cancer therapeutics [1]. Merino et al. conjugated anti-PD-L1 or its Fab’ fragment to liposomes, which showed greater antitumor efficacy, both in vitro and in vivo, against melanoma. Additionally, an increase in effector cytotoxic CD8^+^ T cells was noted [126]. Wang et al. attempted to combine PD-L1 monoclonal antibodies with miR-130a/oxaliplatin-loaded liposomes against gastric cancer. The results indicated that high expression of PD-L1 receptors led to higher cellular uptake and increased efficacy [183]. 

Importantly, not all types of tumors will show a response to ICPs. Due to altered cellular and molecular characteristics of the TME, only inflamed tumors containing infiltrating T cells, a broad chemokine profile, and a type I IFN signature (indicative of innate immune activation) appear to resist immune attack through the dominant inhibitory effects of immune system [184]. 

### 6.3. In Situ Vaccination

In situ vaccination (ISV), also referred to as therapeutic vaccination [185], has received some attention in recent years, particularly when used with lipid nanoparticles. While still a strategy based on vaccination, it does not rely on the isolation of TAAs or neoantigens; rather, it induces expression of antigens on/in the tumor. It operates in the gap of tumor immunity where insufficient antigen presentation increases the burden of disease. It is also personalized, inasmuch as it relies on capture of endogenous tumor antigens.

Chen et al. designed lipidoid nanoparticles which were able to capture endogenous antigens, and simultaneously induce the STING pathway, which enhances the processing and presentation of antigens, as well as translation of pro-inflammatory cytokines including type 1 IFNs [4,186,187,188]. These nanoparticles, named 93-O17S-F/cGAMP, when delivered after initial administration of DOX, showed excellent antitumor activity. Not only did 35% of mice totally recover, but 71% of surviving mice were able to completely recover from a subsequent tumor challenge (Figure 7). This study represents an elegant foray into the “smart” territory of lipid nanoparticle applications. It combines a personalized element (antigenic capture) with immunogenic chemotherapy and ligands of cellular pathways. Notably, even though these nanoparticles were not administered systemically, a systemic effect was achieved [4]. A possible weakness of this method, which appears extremely promising, is the rapidly mutative nature of cancer, which may cause tumor cells to mutate and alter nature, characteristics, presentation of TAAs. This may eliminate the possibility for specific immune memory.

## 7. Application in Clinic

With the development of nanomaterials and a series of inspiring results in preclinical studies, a large number of liposomal drugs have recently appeared in clinical trials for anticancer therapy. Doxil^®^ was the first liposomal drug to be approved by the FDA. Its success has led to many liposomally delivered cytotoxic drugs acquiring application licenses and successfully being commercialized. These drugs, which show activity in a variety of solid and hematological malignancies, include doxorubicin, daunorubicin, cytarabine, vincristine, mifamurtide, irinotecan, and cisplatin [30,189]. 

Alongside these, there are many combination-antibody-conjugated drug-loaded liposomes emerging in clinical trials. For example, MBP-426^®^ is an oxaliplatin-loaded anti-transferrin-conjugated liposome intended for gastroesophageal, gastric, or esophageal adenocarcinomas [30]. MM-302 is conjugated to anti-HER2 antibodies and loaded with doxorubicin, intended for advanced HER2-positive breast cancer [30]. Immunological techniques could also be used, where co-delivery of nucleic acid drugs, such as SGT-53, consisting of anti-transferrin scFvs and p53 plasmids in a cationic liposome, has been explored [159]. These immune targeting liposomes have attracted great attention in recent years; thus, we refer here to several high-quality reviews which give detailed descriptions thereof [30,100,159]. Aside from the above, certain liposomal cancer vaccines which have recently advanced into clinical trials are discussed below and presented in full in Table 4.

### 7.1. Lipo-MERIT (Melanoma FixVac)

FixVac is an intravenously administered liposomal RNA (RNA-LPX) vaccine, which consists of four naked RNA molecules encoding New York Esophageal squamous cell carcinoma 1 (NY-ESO-1), melanoma-associated antigen A3 (MAGE-A3), tyrosinase, and transmembrane phosphatase with tensin homology (TPTE), aiming at TAAs of melanoma. These RNA-lipoplexes target patients with advanced melanoma to evaluate the safety and tolerability in a phase I study. An interim analysis published in the year 2020 shows that FixVac can induce strong CD4^+^ and CD8^+^ T-cell immunity against the vaccine antigens. Also, they observed the patients with anti-PD1 failure responded to rechallenge with anti-PD1 therapy after FixVac monotherapy [190].

### 7.2. DPX-0907

Berinstein et al. developed a novel peptide-based nano vaccine called DPX-0907, using the DepoVaxTM delivery system, which consists of seven tumor-specific HLA-A2-restricted peptides, a universal T helper peptide (modified tetanus toxin peptide), a polynucleotide adjuvant, a liposome, and montanide ISA51 VG. This multifunctional vaccine was proved to be safe, showing limited toxicity, which could stimulate antigen-specific T-cell responses in 61% of patients and secrete multiple type 1 cytokines [86,191,192].

### 7.3. L-BLP25 (Tecemotide)

L-BLP25 is a lyophilized liposomal product consisting of transmembrane glycoprotein mucin 1 (MUC1) peptide, immune-adjuvant MPLA and three lipids (Chol, DMPG, and DPPC). MUC1 is normally found on the surface of secretory epithelial cells, but is overexpressed and aberrantly glycosylated on various cancer cells, and is associated with tumor progression and poor prognosis [193]. Butts et al. reported a phase IIB study in 2005 which evaluated the L-BLP25 in Stage IIIB and IV non-small-cell lung cancer (NSCLC). Patients after first-line treatment were randomly assigned to either L-BLP25 plus best supportive care (BSC) or BSC alone. Initial results and further follow-up from this maintenance treatment suggested a potential survival advantage. An improvement of overall survival (OS) was observed in patients treated with L-BLP25 plus BSC of 4.2 months over those treated with BSC alone. However, only 16 of 78 assessable patients in the vaccine arm developed an immune response [194,195]. A further phase III study which enrolled 1513 patients continued the investigation of L-BLP25 in combination with cyclophosphamide pretreatment in order to reduce the number of Tregs which could suppress antitumor immune responses [196]. Outcomes showed median survival time (MST) was 25.6 months when treated with L-BLP25, versus 22.3 months when treated with placebo. MST in patients who received previous concurrent chemoradiotherapy was 30.8 months in the L-BLP25 group compared with 20.6 months in the control [197]. This novel peptide vaccine was also tested in different ethnicities including Caucasian and East-Asian. However, a clinical trial in Japan did not support the results from the phase III study. No improved MST with L-BLP25 in the subgroup of patients treated with primary concurrent chemoradiotherapy was detected [198]. 

L-BLP25 was also applied in other tumors such as melanoma, prostate cancer, colorectal adenocarcinoma, multiple myeloma (MM), and breast cancer [199]. L-BLP25 showed promise in prolonging the doubling time of prostate-specific antigen in hormone-naive men with biochemical failure after prostatectomy [200]. However, the study for patients who acquired resection of colorectal liver metastases failed to meet its primary endpoints, such as significantly improving relapse-free survival (RFS) and OS with L-BLP25 treatment [201]. Lastly, no improvement for residual cancer burden (RCB) or pathological complete response (pCR) rates were discovered in breast cancer patients receiving standard neoadjuvant therapy [202].

### 7.4. Lipovaxin-MM

Lipovaxin-MM is a multicomponent DC-targeted liposomal vaccine for metastatic melanoma. In a phase I clinical study, a multifunctional metal-chelating liposome platform was able to incorporate a specific antibody fragment into a DC surface receptor (particularly expressed on immature DCs). Tumor antigen peptides including melanocyte differentiation antigens, gp100, tyrosinase, and MelanA/MART-1 were used, alongside IFN-γ. The results showed limited vaccine toxicity and was well tolerated. However, the overall results showed minimal immunogenicity and low correlation to clinical outcomes [78]. The majority of patients enrolled in this experiment had advanced metastatic disease and were elderly. Age-related immuno-senescence [207] or disease-related immune suppression may have made contributions to the weak immune responses observed [208].

### 7.5. PDS0101

PDS0101 is a lipid nanoparticle vaccine which contains two active components. The first is called R-DOTAP (R-1,2-dioleoyl-3-trimethylammonium-propane; Versamune), the immunologically superior R-enantiomer of the cationic lipid DOTAP. R-DOTAP has been shown to activate bone marrow-derived DCs and stimulate a cytokine response in vivo [209]. The second group of active components are six human leukocyte antigen (HLA)-unrestricted epitopes against HPV16 E6 and E7, which have been proven to elicit antigen-specific CD4^+^ and CD8^+^ T cells in a phase I clinical study (NCT02065973). In preclinical research, the PDS0101 vaccine can generate HPV-specific T-cell immune responses and antitumor activity in an animal model for cervical cancer. Maximum antitumor effects were observed when co-administered with immune modulators (Bintrafusp Alfa and NHS-IL-12) [210]. Due to these positive results, a phase IIA clinical trial was designed to clarify the effect of PDS0101 when given together with chemotherapy and radiation therapy in stage IB3-IVA cervical cancer. It is currently undergoing recruitment and the estimated primary completion date is the year 2024 (NCT04580771).

### 7.6. Autologous Tumor Cell Vaccine

Therapeutic vaccination with unique determinants of the variable regions of clonal tumor immunoglobulin molecules are termed idiotypes (Id). These are often expensive and labor intensive to produce. Neelapu et al. extracted membrane proteins from lymph node biopsies of each patient and incorporated these directly into liposomes, along with IL-2 to produce membrane-patched proteoliposomes, which can be produced within one day. Each vaccine was formulated on a per milliliter basis with membrane proteins obtained from approximately 1.6 × 10^8^ biopsy cells, 4 × 10^6^ IU of IL-2, and DMPC liposomes. In a phase I study, 20 patients with stage III and IV follicular lymphoma were enrolled. The results showed good tolerance. Post-vaccine peripheral blood mononuclear cells (PBMCs) from 5 out of 10 patients responded to autologous tumor cells by producing IFN-γ, granulocyte-macrophage colony-stimulating factor (GM-CSF), and/or TNF-α. Humoral anti-Id responses were not detected in any evaluated patients. Therefore, this novel formulation may serve as a model for vaccine development against other human malignancies where tumor-associated antigens have not been defined [203,204].

### 7.7. dHER2 + AS15 Vaccine

A recombinant HER2 protein, including a truncated intracellular domain and the complete extracellular domain, combined with the immunological adjuvant AS15 which consists of 3-O-desacyl-4′-monophosphoryl lipid A (monophosphoryl lipid A; MPL), QS-21 (Quillaja saponaria extract) and a synthetic oligodeoxynucleotide containing unmethylated CG dinucleotides (CpG ODNs 7909), in a liposomal formulation. In two early phase clinical studies of patients with HER2-overexpressing breast cancer, the data showed these immunizations were safe and the clinical outcomes were associated with antibody and T-cell responses [205].

### 7.8. ONT-10

ONT-10 is a therapeutic peptide vaccine which contains synthetic MUC1 antigens (M40Tn6) and novel synthetic TLR-4 agonists (PET lipid A) in a liposomal formulation. A phase I clinical study was designed for patients with incurable solid tumors expressing MUC1. A poster presentation by Nemunaitis showed that MUC1-specific antibody responses could be seen in the majority of patients and stable disease (SD) period of more than 6 months was seen in 28% of patients [206].

More liposomal vaccine clinical trials are undergoing recruiting. W_ova1 is a liposomal mRNA vaccine containing three ovarian cancer TAA-encoding RNAs. A phase I clinical study in combination with chemotherapy, in ovarian cancer patients is in progress (NCT04163094). A first-in-human phase I study of liposomal RNA vaccines for newly diagnosed adult O6-methylguanine-DNA methyltransferase (MGMT) unmethylated glioblastoma (GBM) have recently begun recruiting. Autologous total tumor mRNA and liposomal (DOTAP) pp65 full-length lysosomal-associated membrane protein (LAMP)-mRNA-loaded vaccines were designed to incorporate with surgery and radiation therapy. Potentially eligible participants will be enrolled for sterile collection of tumor material in a manner suitable for RNA extraction, amplification, and loading of lipid particles (LPs) (NCT04573140) [211].

## 8. Limitations and Future Prospects

### 8.1. General Areas of Improvement

Lipid-based nanoparticles are defined by their multifunctionality and tailorability for the diverse delivery capabilities thereof. Abundant research in preclinical and clinical studies illustrates the potential for lipid-based applications in the future. Immunoliposomes in combination with other agents, including cytotoxic drugs and nucleic acids, are able to directly target signaling pathways for tumor survival, or to co-deliver antitumor drugs. With development in the understanding of antitumor immune responses, novel IILFs have been emerging rapidly in recent years. These include liposomal cancer vaccines for APC or other immune cell targeting. This concept shifts toward “immune modulation”, which is promising for cancer therapy in the years to come.

Aside from the above, liposomal nanoparticles exhibit more precise targeting abilities and stronger immune responses when exposed to external stimuli, such as hyperthermia, ultrasound, radiation, and magnetic fields, via encapsulation of specific agents [156,212,213,214,215]. This is in accordance with the term, “Chemo Bomb”, coined by Dr ten Hagen in 2010, to describe triggered drug release from liposomes [216]. Also, some advanced gene modification methods, such as CRISPR/Cas9, can be delivered by liposomes, which can boost antitumor immune responses by editing genes related to tumor progression. Moreover, several novel administration techniques, like microfluidic devices, can help to target specific sites like lymph nodes, directly [217]. Meanwhile, the capacity of liposomally-packaged imaging agents can serve as real-time monitoring systems for both diagnostic and therapeutic purposes [179].

Theoretically, liposomes could carry any number of combinational therapies with suitable manufacturing techniques. However, the still-poorly understood nature of synergistic or antagonistic interactions between therapies according to their intrinsic mechanisms mean that expectations of a “1 + 1 > 2” result from random combinations may be premature. Combination of compounds on or in liposomes allows the setting of optimal ratios at which synergy occurs. It is known that synergy between two or more compounds is dependent on location and concentrations in space and time; thus, stable liposome-mediated packaging and delivery of compounds to a tumor guarantees the simultaneous delivery of these compounds at desired ratios, and facilitates the aforementioned space- and time-controlled delivery. Additionally, it is difficult to evaluate the mutual effects and side-effects, since moieties coupled in liposomes are complex and diverse. Importantly, combination of multiple therapeutics complicates clinical approval and will most likely result in unacceptable production costs. Due to the combinational applications of several antigens, adjuvants, and repeated administration strategies, irAEs, in particular, should be taken into consideration. 

Even with these preclinical advances in liposomal nanoparticle immune therapies, there are, at the time of writing, still none which have been approved by the FDA. There are a number of reasons for this. Firstly, systematic parallel screening of the myriad of IILFs’ properties remains difficult, owing to the challenge of rapid, precise, and reproducible synthesis. Exacerbating this is that antitumor effects of, and immune responses to, IILFs are highly variable across tumor type and location. Furthermore, conventional in vitro models using 2-dimensional cell culture techniques lack the complexity of biological tissues, which may fail to capture the intricate interplay of IILFs with physiological barriers and complex, interdependent systems. Lastly, the great diversity of IILFs makes manufacturing a significant challenge from preclinical to clinical development, and subsequently, commercialization [24]. 

### 8.2. Personalized Medicine Approaches

As discussed in the relevant sections of this paper, IILFs present a promising platform upon which personalized medicine approaches can be built. In situ vaccination, for example, makes use of antigen capture mechanisms for presentation of highly personalized endogenous antigens [4]. This has also been attempted using antigen extraction and isolation from patient samples, which has been less effective, given the dynamic nature of tumors and changes in antigen presentation [17,204]. Sophisticated methods of achieving such outcomes, including immunopeptidomics and neoantigen-reactive T-cell detection, are discussed in reviews by Chong et al. [218] and Yossef et al. [219]. Patient stratification via immunophenotyping and immune profiling can also aid in prediction of a desired response to treatment, especially in immunotherapy [220].

On the other hand, the EPR effect remains stochastic and poorly defined [221], but has an important effect on liposome pharmacokinetics [143,152]. The targeting and leveraging of the EPR effect for treatment as discussed in this paper would strongly benefit from patient-to-patient characterization thereof prior to treatment. This may be done with imaging agents of similar characteristics to intended liposomal treatment [144,222]. Nagamitsu et al. and Ding et al. have reported feasible methods for imaging the EPR effect which may be employed more widely in future [223,224]. 

Of course, anything personalized in medicine becomes expensive, time-consuming, labor intensive, and thereby, less feasible. However, with proven benefit and an emphasis on clinical application and scalability in current preclinical nanomedical research, a personalized medicine approach may become an important, accessible and necessary facet of nanomedical modalities in the near future. 

## 9. Conclusions

In conclusion, multidisciplinary therapy is the inevitable trend for tumor treatment in the future. In order to overcome such a formidable enemy, a smart lipid-based nanoplatform having specific antibody loading, controllable drug release and the ability to be monitored non-invasively may become the ideal weapon, according to differing individual immunophenotyping, tumor antigen expression, resistance to chemotherapy and health economy conditions.

## Figures and Tables

**Figure 1 pharmaceutics-14-00026-f001:**
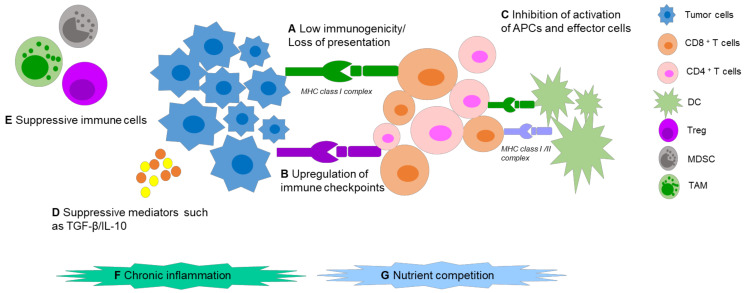
Mechanisms of tumor immune escape. (**A**) Tumor-associated antigens show low immunogenicity and/or develop several antigen presentation problems. (**B**) Upregulation of immune checkpoints like PD-1 on T-cell surface or its ligand PD-L1 on tumor surface can cause T-cell anergy and inhibit the antitumor immune response. (**C**) Low secretion of immune stimulatory molecules in the TME, like IL-2, which inhibits the activation of APCs and other effector cells such as CD8^+^/CD4^+^ T cells. (**D**,**E**) Production of immunosuppressive cytokines by tumor cells, such as TGF-β and IL-10 and recruitment of several immunosuppressive cells like regulatory T cells (Tregs), Tumor-associated macrophages (TAMs) and myeloid-derived suppressor cells (MDSCs), are able to attenuate antitumor immune response. (**F**,**G**) Chronic inflammation and nutrient competition promote immune dysfunction. Abbreviations: MHC (major histocompatibility complex); CD (cluster of differentiation).

**Figure 2 pharmaceutics-14-00026-f002:**
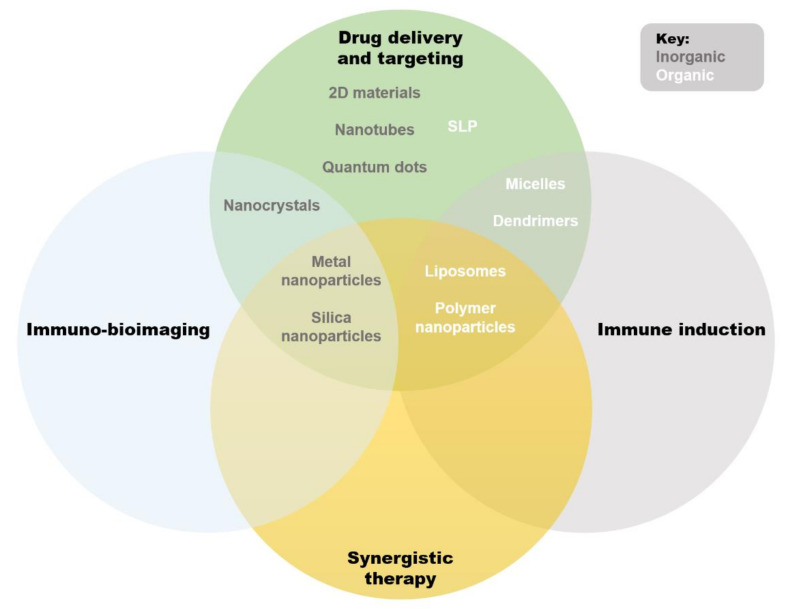
Qualitative Venn diagram showing an overview of primary applications and overlaps of available nanosystems used in immunotherapy. Abbreviations: SLNs (solid lipid nanoparticles).

**Figure 3 pharmaceutics-14-00026-f003:**
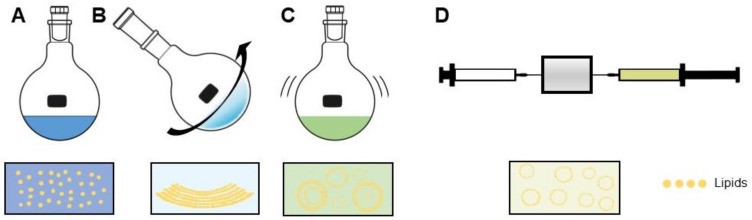
Liposome synthesis using the thin-film hydration method. (**A**) Lipids are combined in desired ratios in organic solvent. (**B**) The solution then undergoes rotary evaporation (dehydration) under negative pressure, causing formation of lipid bilayers. (**C**) The dry lipid film is rehydrated in aqueous buffer with swirling and sonication, which leads to non-uniform formation of MLVs, LUVs and GUVs. (**D**) Finally, extrusion through membranes of sequentially lower pore size forces formation of SUVs within a small size range, having uniform characteristics.

**Figure 4 pharmaceutics-14-00026-f004:**
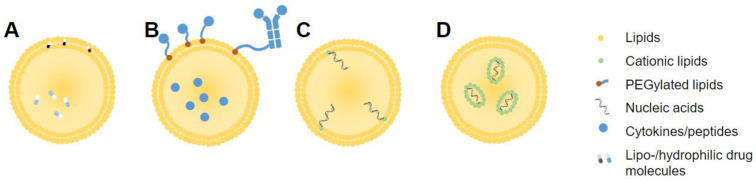
Simplified locations of incorporated or conjugated agents, including (**A**) drug molecules, (**B**) cytokines/peptides, or (**C**) nucleic acids in liposome or (**D**) solid lipid nanoparticle formulations.

**Figure 5 pharmaceutics-14-00026-f005:**
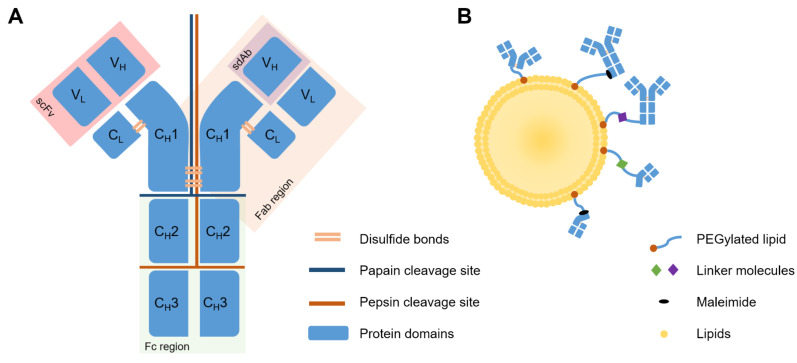
(**A**) Basic structure and enzymatic cleavage sites of a typical IgG antibody, with (**B**) conjugation methods and schematic linking sites to liposomes. Abbreviations: Fc (fragment crystalline); Fab (fragment antigen-binding); C (conserved); V (variable); H (heavy); L (light); scFv (single-chain variable fragment); sdAb (single-domain antibody); PEG (polyethylene glycol).

**Figure 6 pharmaceutics-14-00026-f006:**
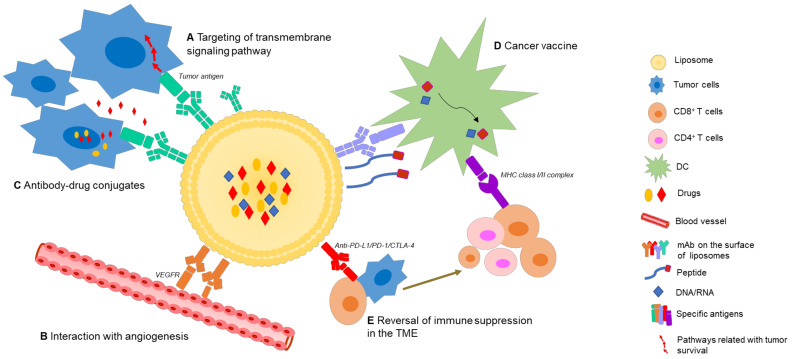
Therapeutic strategies of immunoliposomes and IILFs. (1) Antibody-conjugated liposomes: (**A**) directly inhibit tumor transmembrane signaling pathways related to survival by binding the specific antigens on tumor cells, (**B**) interact with the TME (for example, targeting angiogenesis) which delivers essential nutrients to the tumor, and (**C**) kill tumor cells by co-delivering drugs, including cytotoxic chemical drugs. (2) Immune-inducing liposomal formulations: (**D**) present antigens and/or adjuvants to APCs and stimulate an antitumor immune response and (**E**) block immune suppressive molecules on tumor cells (PD-L1) or immune cells (PD-1/CTLA-4). Abbreviations: TME (tumor microenvironment); PD (programmed death); mAb (monoclonal antibody).

**Figure 7 pharmaceutics-14-00026-f007:**
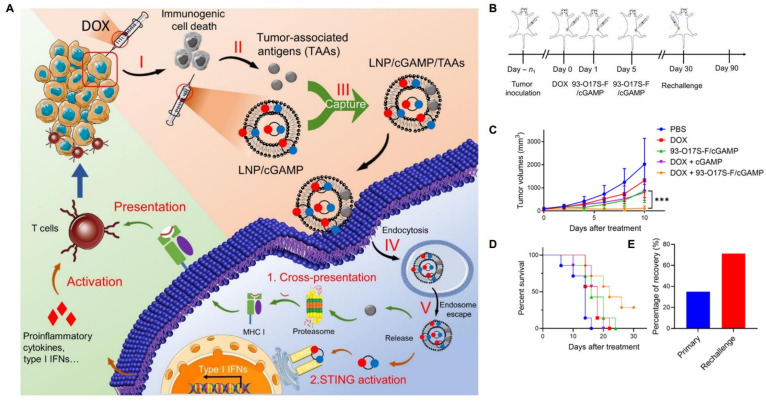
(**A**) Study design and description of in situ vaccination experiments using lipid nanoparticles. (**B**) Mice were treated on Day 0 with DOX, then on Days 1 and 5, with the LNP, 93-O17S-F/cGAMP. On Day 30, mice were rechallenged with tumor cells of the same kind. (**C**) Tumor growth was significantly (*p* ≤ 0.001) reduced with 93-O17S-F/cGAMP treatment during the first 10 days after treatment. (**D**,**E**) Mouse survival pre- and post-rechallenge, up until 30 days, and thereafter. Abbreviations: PBS (Phosphate-buffered saline); DOX (doxorubicin); cGAMP (cyclic guanosine monophosphate-adenosine monophosphate); 93-O17S-F/cGAMP (the specific lipidoid nanoparticle used in this study); LNP (lipid nanoparticle); MHC (major histocompatibility complex); STING (stimulator of interferon genes); IFN (interferon). Reprinted/adapted from Chen et al. [4]. © The Authors, some rights reserved; exclusive licensee AAAS. Adapted from http://creativecommons.org/licenses/by-nc/4.0/ (accessed on 17 December 2021). These figures have been modified for exclusion of irrelevant detail in the context of this review.

**Table 1 pharmaceutics-14-00026-t001:** Advantages and disadvantages of liposomes.

Advantage	Explanation	References
Biocompatibility	Non-toxic, biodegradable, non-immunogenic	[30]
Amphiphilicity	High solubility of various compounds (both hydrophobic and hydrophilic, and compatible with many physiological cavities)	
Smart	Stimuli-responsive and multifunctional	[8]
Easy, accessible formulation procedures	With simple and inexpensive equipment, liposomes can be synthesized and customized in any laboratory	[10]
Rational design, customization is an option (flexibility)	For example, attaching fluorophores, polyethylene glycosylation (PEGylation)	
Drug packaging/protection	Reduced toxicity and clearance of encapsulated agent, controlled release Reduced unwanted exposure to normal tissues	[8]
Applicable in multiple therapies	Including oncology, infectious diseases, and vaccinations	[34,35,36,37,38]
Ab conjugation and other functionalization possibilities	Active targeting Therapeutic immune induction	[8]
Deeper penetration to physiological compartments	e.g., blood–brain tumor barrier, deeper tissues	[39]
**Disadvantage**		
Lack of specificity in delivery applications	The enhanced permeation and retention effect is often relied upon, with varying success	[9]
Stability/half-life	Low stability can lead to leakage of encapsulated drugs or premature degradation	[8]
Unreliable drug packaging based on low loading efficiency and drug leakage	In some cases loading efficiency and leakage can prevent therapeutic progression	[40]
Certain cell layers not transversable	Stratum corneum cannot be crossed, blood–brain barrier can only be crossed with modifications (still being tested)	[41,42]
Rigidity	Can cause insufficient drug release	[30,41]
Upscaling challenges	Scale up requires stringent quality control and is labor intensive	[8]

**Table 2 pharmaceutics-14-00026-t002:** Lipids and other agents commonly used in the formulation of liposomes.

Type of Lipid or Agent	Application	Lipids in Use | Tm | Charge at Neutral pH	References
Saturated phospholipids	Tight packing of lipids for formation of a stable bilayer due to straight acyl chains Rigidity and stability Reduce clearance by the mononuclear phagocytic system Certain lipids can be modified for increased stability under certain conditions or for certain purposes	1,2-Distearoyl-sn-glycero-3-phosphocholine (DSPC) | 55 °C | Neutral	[43,44,45,46,47,48,49,50,51,52]
		1,2-Dipalmitoyl-sn-glycero-3-phosphocholine (DPPC) | 41 °C | Neutral	
		1,2-Dipalmitoyl-sn-glycero-3-phosphoglycerol (DPPG) | 41 °C | Neutral	
		1,2-Distearoyl-sn-glycero-3-phosphoethanolamine (DSPE) | 74 °C | Neutral	
		1,2-Dilauroyl-sn-glycero-3-phosphocholine (DLPC) | −2 °C | Neutral	
		1,2-Dimyristoyl-sn-glycero-3-phosphocholine (DMPC) | 24 °C | Neutral	
		1,2-Dimyristoyl-sn-glycero-3-phosphoglycerol (DMPG) | 23 °C | Negative	
		1,2-Diphytanoyl-sn-glycero-3-phosphocholine (DPhPC) | -- * | Neutral	
		1,2-Dimyristoyl-sn-glycero-3-phosphoethanolamine (DMPE) | 50 °C | Neutral	
		1,2-Dilauroyl-sn-glycero-3-phosphorylethanolamine (DLPE) | 29 °C | Neutral	
		1,2-Dipalmitoyl-sn-glycero-3-phosphoethanolamine (DPPE) | 60 °C | Neutral	
		1,2-Dipalmitoyl-sn-glycero-3-phosphoethanolamine (DPPE) | 60 °C | Neutral	
		1,2-Dilauroyl-sn-glycero-3-phosphoglycerol (DLPG) | −3 °C |Negative	
		1,2-Distearoyl-sn-glycero-3-phosphoglycerol (DSPG) | 55 °C |Negative	
		1,2-Dipalmitoyl-sn-glycero-3-phospho-L-serine (DPPS) | 51 °C |Negative	
		1,2-Distearoyl-sn-glycero-3-phospho-L-serine (DSPS) | 68 °C |Negative	
		1,2-Dilauroyl-sn-glycero-3-phosphate (DLPA) | 31 °C |Negative	
		1,2-Dimyristoyl-sn-glycero-3-phosphate (DMPA) | 52 °C |Negative	
		1,2-Dipalmitoyl-sn-glycero-3-phosphate (DPPA) | 65 °C |Negative	
		1,2-Distearoyl-sn-glycero-3-phosphate (DSPA) | 75 °C |Negative	
		1,2-Dioleoyl-sn-glycero-3-phosphate (DOPA) | −4 °C |Negative	
		1,2-Dimyristoyl-sn-glycero-3-phospho-L-serine (DMPS) | 35 °C | Negative	
		1,2-Dilauroyl-sn-glycero-3-phospho-L-serine (DLPS) | 30 °C | Negative	
		1-Palmitoyl-2-hydroxy-sn-glycero-3-phosphocholine (lyso-PPC) | 42 °C |Neutral	
Natural phospholipids	Cost-effective Soybean PC (lecithin): stable and retains content suitably Other sources include sunflower and rape seeds, chicken egg yolk, and porcine or bovine brain Can be a blend of lipid molecules, so exact T_m_ cannot be determined	Hydrogenated soy l-α-phosphatidylcholine (HSPC) | Neutral	[44,46,53,54,55,56]
		L-α-phosphatidylcholine (Egg PC) | Neutral	
		3-sn-phosphatidyl-L-serine (Brain PS) | Negative	
		L-α-Phosphatidylglycerol (Egg PG) | Negative	
Unsaturated phospholipids	Tune bilayer fluidity at physiological temperatures Allow formation of lipid domains (rafts) Rafts facilitate embedding of functional moieties Choice between neutral and negatively charged lipids for pharmacokinetic tuning of liposomes Negatively charged (POPS) lipids increase interaction with endothelial and tumor cells, and thus clearance	1-Palmitoyl-2-oleoyl-sn-glycero-3-phosphoglycerol (POPG) | −2 °C | Negative	[28,30,46,49,53,57,58,59,60,61,62,63,64]
		1-Palmitoyl-2-oleoyl-sn-glycero-3-phosphocholine (POPC) |−2 °C | Neutral	
		1-Palmitoyl-2-oleoyl-sn-glycero-3-phospho-L-serine (POPE) | 25 °C | Neutral	
		1,2-Dioleoyl-sn-glycero-3-phosphocholine (DOPC) | −17 °C | Neutral	
		1-Palmitoyl-2-oleoyl-sn-glycero-3-phospho-L-serine (POPS) | 14 °C | Negative	
		1,2-Dioleoyl-sn-glycero-3-phospho-L-serine (DOPS) | −11 °C | Negative	
		1,2-Dioleoyl-sn-glycero-3-phosphoethanolamine (DOPE) | −16 °C | Neutral	
		1,2-Dioleoyl-sn-glycero-3-[phospho-rac-(1-glycerol)] (DOPG) | −18 °C | Negative	
		Sphingomyelin (SM) | 37 °C | Neutral	
		1-Stearoyl-2-arachidonoyl-sn-glycero-3-phosphoethanolamine (SAPE) | nt ** | Neutral	
		1,2-Dioleoyl-sn-glycero-3-ethylphosphocholine (EPC) | −16 °C | Positive	
		1,2-Dioleoyl-3-trimethylammonium-propane (DOTAP) | <5 °C | Positive	
		1,2-Stearoyl-3-trimethylammonium-propane (DSTAP) | 63 °C | Positive	
		1,2-Di-O-octadecenyl-3-trimethylammonium propane (DOTMA) | nt ** | Positive	
		1,2-Dipalmitoyl-3-trimethylammonium-propane (DPTAP) |41 °C | Positive	
		1,2-Dimyristoyl-3-trimethylammonium-propane (DMTAP) | 24 °C | Positive	
		1,2-Dioleyloxy-3-dimethylaminopropane (DODMA) | nt ** | Positive	
PEGylated phospholipids	Adds a stealth element to liposomes (5–10%) Prevent aggregation and prolong circulation Length of PEG can be specified according to need PEG-maleimide functionalization is used for ligand coupling at the distal end of the PEG chain (maximum visibility) Can assist state-change and content-release at transition temperature (Tm) in the case of stimuli-responsive liposomes	1,2-Distearoyl-sn-glycero-3-phosphoethanolamine-N-(amino (polyethylene glycol)-2000) (DSPE-PEG(2000))	[7,44,65,66,67]
		1,2-Distearoyl-sn-glycero-3-phosphoethanolamine-N-(amino (polyethylene glycol)-5000) (DSPE-PEG(5000))	
		DMPE-((polyethylene glycol)-2000) (C14-PEG)	
		1,2-Dimyristoyl-rac-glycero-3-methoxypolyethylene glycol-2000 (DMG-PEG(2000))	
Cholesterols	Maintain phase and integrity of bilayer No effect on release of contents (without intervention) Buffers temperature’s effects on bilayer integrity Ratio can be adjusted for desired membrane integrity Cholesterol can be used as a site for embedding of molecules or proteins	Cholesterol (Ch)	[46,57,58,68]
		Cholesteryl hemisuccinate (CHEMS)	
		3β-[N-(N’,N’-Dimethylaminoethane)-carbamoyl]cholesterol (DC-Chol)	
Non-lipid additives	Can add valuable, unique or desirable characteristics to particles Can be known as lipidoids e.g., Span 85 adds stability to emulsions	Squalene	[62,67,69,70]
		C12-200	
		Dimethyldioctadecylammonium (DDA)	
		Sorbitane trioleate (Span 85)	
		Diacetyl phosphate (DCP)	
Internal milieu of nanoparticles	Buffers with pH, tonicity, and appropriate solvent characteristics desired for a particular purpose, be it drug packaging, biological stability, or solute compatibility	Wide range of common buffers used successfully	[33]
Labels and dyes	Fluorescently labelled lipids, denoted by “--/” (0.01 –0.03%) for membrane bilayer incorporation or strong association hitherto Labels can be combined to suit the needs and limitations of equipment	7-Nitro-2-1,3-benzoxadiazol-4-yl--/ (NBD--lipid)	[45,71,72]
		/--Lissamine rhodamine B sulfonyl) (Liss-rhod--lipid)	
		Vybrant™ DiD	
		3,3′-Dioctadecyloxacarbocyanine Perchlorate (DiO)	
		1,2-Dioleoyl-sn-glycero-3-phosphoethanolamine-N (TopFluor^®^ AF488) (Various wavelengths available)	
		Texas Red-1,2-dipalmitoyl-sn-glycero-3-phosphoethanolamine (TR-DPPE)	
Ligands, targeting, and immunogenic molecules	Small ligands are preferred, not whole antibodies (antigen-binding fragments, single-chain variable fragments, nanobodies, and peptides) for avoidance of immunogenicity and steric hindrance These can be with a targeting purpose, or an immune modulation purpose, such as ICB, T-cell interaction or cytokine delivery Chemotherapeutics packaged internally for hydrophilic drugs, and within lipid bilayers for lipophilic drugs Certain lipids are also functional as vaccine adjuvants (QS-21, DDAB, and MPLA) Others enhance affinity of lipid bilayers for certain markers on proteins (CoPoP interacts with his-tags) Others possess chelation ability (3NTA-DTDA)	Quillaja saponaria extract (QS-21)	[7,40,44,65,73,74,75,76,77,78,79]
		Dimethyldioctadecylammonium (DDAB)	
		Monophosphoryl lipid A (MPLA)	
		Cobalt porphyrin phospholipid (CoPoP)	
		3(Nitrilotriacetic acid)-ditetradecylamine (3NTA-DTDA)	

Abbreviations: T_m_, transition temperature; PEGylated, polyethylene glycolated. * DPhPC does not exhibit a phase change as tested between −120 and 120 °C. ** Nt = not tested.

**Table 3 pharmaceutics-14-00026-t003:** Notable applications of cancer vaccines in the last five years.

VACCINE TYPE		Antigenic Loading	Deriving	Co-Administration Payload	LP Formulation	Tumor Model	Efficacy *	References
Peptide	E7_49–57_		HPV 16 E7	CoPoP, PHAD, QS-21	DOPC/Cholesterol	TC-1 cell line	①②③⑤	[169]
				CpG ODN	SPC/Cholesterol/DOTAP, DSPE-PEG(2000), DSPE-PEG(2000)-Mannose		①②③⑤	[170]
	HPV16 E7 oncogene		HPV16 E7	A helper peptide from influenza virus, A synthetic agonist (Pam_2_CAG or MPLA/C12iEDAP)	PC/PG/Cholesterol/DPPG-Maleimide	TC-1 cell line	①②③④	[76,168]
	mn		Integrin αvβ3	AL3810	HSPC/Cholesterol/mPEG(2000)-DSPE, HSPC/Cholesterol/mPEG(2000)-DSPE/PEG3400-DSPE	Glioma	①⑤	[171]
	P5 + P435		HER2/neu		DSPE-PEG(2000)-Maleimide, DSPC/DSPG/Cholesterol/DOPE	Breast cancer	①②③	[166]
	BCMA_72–80_		B cell		Cholesterol/DOPC/DOTAP	Multiple myeloma	②③④	[172]
	ADPGK		Neoantigen from mutation	Black phosphorus quantum dots		Colorectal cancer	①②③④	[173]
	gp100		TAA	CpG-ODN	DSPE-PEG(2000)-Maleimide, DOTAP/Cholesterol	Melanoma	①②③④	[174,175]
	TRP2_180-188_		TAA	CpG-ODN	POPC/Cholesterol/DSPE-PEG2000/DDAB/mannose lipid, DSPE-PEG(2000)-Maleimide	Melanoma	①②③	[79]
	P5		HER2/neu		DSPC/DSPG/Cholesterol	TUBO in vivo tumor mice model	①②③	[176]
	Synthetic LHRH		TAA	Tetanus toxoid T-helper epitope_830–844_, several TLR ligands	EPC/PG/palmitoyl, DSPE-PEG(3400)-Maleimide	Androgen-responsive prostate cancer	③	[177]
RNA	Total RNA		Liver cancer cells		DOTAP	Hepato-cellular carcinoma	①②④	[178]
	OVA			Iron oxide	DOTAP/Cholesterol	Melanoma	①③④⑤	[179]
	gp70 RNA		Moloney murine leukemia virus		DOTMA/DOPE	BALB/c mice	②④⑤	[180]
DNA	Individual somatic mutations using WES				DOTAP/DOPE	Melanoma	①②④	[181]

Abbreviations: CoPoP (cobalt-porphyrin-phospholipid); PHAD (a synthetic derivative of MPLA); QS-21 (Quillaja saponaria extract); CpG ODN (CpG oligodeoxynucleotides); DOPC (1,2-dioleoyl-sn-glycero-3-phosphocholine); SPC (1-stearoyl-sn-glycero-3-phosphocholine); DOTAP (1,2-dioleoyl-3-trimethylammonium-propane); DSPE (1,2-Distearoyl-sn-glycero-3-phosphoethanolamine); PEG (polyethylene glycol); TC-1 (E7-transformed tumor cells derived from murine pulmonary epithelial cells); Pam2CAG (dipalmitoyl-cysteine-alanyl-glycine); MPLA (monophosphoryl lipid A); C12iEDAP (acylated derivative of the dipeptide γ-D-Glu-mDAP); PC (phosphatidyl choline); PG (phosphatidyl glycerol); DPPG (1,2-dipalmitoyl-sn-glycero-3-phosphoglycerol); mn (a linear pentapeptide); AL3810 (reversible ATP-competitive inhibitor aiming at the ATP pocket on intracellular region of VEGFR and FGFR); HSPC (Hydrogenated soy l-α-phosphatidylcholine); HER2 (The human epidermal growth factor receptor 2); DSPC (1,2-Distearoyl-sn-glycero-3-phosphocholine); DSPG (1,2-Distearoyl-sn-glycero-3-phosphoglycerol); DOPE (1,2-dioleoyl-sn-glycero-3-phosphoethanolamine); BCMA_72–80_ (a member of the TNF receptor superfamily and the receptor for binding of B cell activating factor and the proliferation-inducing ligand); gp100/TRP2_180–188_ (melanocyte differentiation antigen); POPC (1-palmitoyl-2-oleoyl-sn-glycero-3-phosphocholine); DDAB (dimethyl dioctadecyl ammonium bromide); TUBO cells (a cloned line established in vitro from a BALB-neu T mouse mammary carcinoma); LHRH (Luteinizing-hormone-releasing hormone); TLR (Toll-like receptor); EPC (1,2-dioleoyl-sn-glycero-3-ethylphosphocholine); OVA (B16F10-OVA is a murine melanoma cell line expressing the chicken ovalbumin gene); gp70 RNA (Moloney murine leukemia virus); WES (whole-exome sequencing). * Efficacy of vaccinations is shown according to the following binary (present/not present) criteria: ① Tumor shrinkage; ② Stimulation of tumor-specific T cells; ③ Induction in cytokine production; ④ Induction of APCs or DCs; ⑤ Other factors (including increased numbers of immunoinhibitory cells such as MDSCs, Tregs and macrophages [169,170]; complement effect [171], also NK cells [180] and B cells (for humoral immunity and antibody production) [171], and lymph node or spleen enlargement and increased function [179,180].

**Table 4 pharmaceutics-14-00026-t004:** Application of liposomal cancer vaccines in clinic.

Vaccine	Tumor Type	Trial Phase	Formulation	Identifier	References
FixVac	Melanoma	Phase 1	(1) Four naked RNA molecules, encoding NY-ESO-1, MAGE-A3, tyrosinase, and TPTE, aiming at TAAs of melanoma (2) Liposome	NCT02410733	[190]
DPX-0907	Ovarian, breast prostate cancer	Phase 1	(1) Seven tumor-specific HLA-A2-restricted peptides (P4, P5, P7, P13, P14, P15 and P3)(2) Universal T Helper peptide(modified tetanus toxin peptide)(3) Polynucleotide adjuvant(4) Montanide ISA51 VG(5) Liposomes	NCT01095848	[86,191,192]
L-BLP25 (tecemotide)	NSCLC, Breast cancer, Multiple myeloma, Prostate cancer	Phases 1–3	(1) Synthetic 25 amino acid lipopeptide derived from thetandem repeat region of MUC1 (2) Non-specific adjuvant monophosphoryl lipid A (3) Three different lipids	NCT00960115 NCT01015443 NCT00157196 NCT00157209 NCT01094548 NCT01496131 NCT00925548 NCT00409188 NCT00828009 EudraCT number2011-000218-20	[193,194,195,196,197,198,199,200,201,202]
Lipovaxin-MM	Colorectal adenocarcinoma	Phase 1	(1) A specific antibody fragment to DC surface receptor (2) Tumor antigens including gp100, tyrosinase, and melanA/MART-1 (3) Cytokines IFN-γ (4) 3NTA-DTDA/POPC	NCT01052142	[78]
PDS0101	Cervical cancer	Phase 1/2A	(1) R-DOTAP (2) Six HLA-unrestricted epitopes against HPV16 E6 and E7	NCT02065973 NCT04580771	
Autologous tumor vaccine	Follicular lymphoma	Phase 1	(1) Membrane proteins from autologous lymph node biopsy (2) IL-2 (3) DMPC membrane-patched proteoliposomes	NCT00020462	[203,204]
DHER2 + AS15 vaccine	Breast cancer	Phase 1/2	(1) A recombinant HER2 protein (2) Adjuvant AS15 which consists of MPL, QS-21 and a synthetic oligodeoxynucleotide containing unmethylated CG dinucleotides (CpG ODNs 7909) (3) Liposome	NCT00952692	[205]
ONT-10	Previously treated advanced solid tumors	Phase 1	(1) Synthetic glycolipopeptide MUC1 antigen (M40Tn6) (2) Synthetic TLR-4 agonist (PET lipid A) (3) Liposome	NCT01556789	[206]
W_ova1	Ovarian cancer	Phase 1	(1) Three TAA RNAs (2) Liposome	NCT04163094	
RNA-LP vaccine	Glioblastoma	Phase 1	(1) Autologous total tumor mRNA, (2) pp65 full length LAMP mRNA, (3) DOTAP	NCT04573140	

Abbreviations: NY-ESO-1 (New York Esophageal squamous cell carcinoma 1); MAGE-A3 (melanoma-associated antigen A3); TPTE (transmembrane phosphatase with tensin homology); HLA (human leukocyte antigen); P3 (BAP31); P4 (topoisomerase IIa (TOP2A)); P5 (Integrinβ8 subunit precursor); P7 (Abl binding protein C3); P13 (TACE/ADAM-17); P14 (Junction plakoglobin); P15 (EDDR1); NSCLC (non-small-cell lung cancer); MUC1 (glycoprotein mucin 1); gp100 (melanocyte differentiation antigen); melanA/MART-1 (melanoma antigen recognized by T cells 1); 3NTA-DTDA (3(nitrilotriacetic acid)ditetradecylamine); POPC (1-Palmitoyl-2-oleoyl-sn-glycero-3-phosphocholine); R-DOTAP (R-1,2-dioleoyl-3-trimethylammonium-propane); DMPC (dimyristoylphosphatidylcholine); MPL (3-O-desacyl-4′-monophosphoryl lipid A); QS-21 (Quillaja saponaria extract); CpG ODN (CpG oligodeoxynucleotides); LAMP (lysosomal-associated membrane protein).

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
