# Peer review of "Smart Lipid-Based Nanosystems for Therapeutic Immune Induction against Cancers: Perspectives and Outlooks"

_pharmaceutics, 2021, doi:10.3390/pharmaceutics14010026_

Round 1
Reviewer 1 Report
In this review, Fobian et at discuss lipid-based nanosystems in the setting of cancer therapy, with a particular emphasis on immune oncology. Overall their review is thoughtful and interesting, and my comments minor.
Specific Comments:
-The authors claims that "Cancer immunotherapy has, in many ways, lain dormant since its inception in 1892, owing to poor clinical outcomes; hence tentative adoption in the clinic. Recent years have seen immunotherapy gaining ground, along with plausibility and translationality [3]."
This is incredibly misleading to readers. The last 10 years have seen a rapid advance for cancer immunotherapy, now the preferred treatment for any number of cancer histologies.
Immunotherapy is not merely "gaining ground" in cancer treatment, but is a cornerstone in oncology clinics worldwide. So much so that the identification of immune checkpoints earned the nobel prize in 2018, and we offer our patients immunotherapy on what seems to be a daily basis with zero hesitation.
-Not all figures are mentioned in the text, which needs to be addressed on resubmission.
-Regarding Figure 7, the inclusion of what appears to be already published data is unnecessary and unconventional for a review article. Instead I suggest the authors merely reference this work.
Minor Comments:
-I recommend avoiding the abbreviation IL for immunoliposome, as in immune circles this is associated with "interleukin".
-The authors may want to include a brief discussion of Nal-Irninotecan on page 16 or elsewhere to reinforce the clear promise of liposomal therapies in the clinic. This marks one of the most recent and highest profile success stories for liposomal therapies, and has now been approved as a second line treatment for pancreatic cancer patients who progress on Gemcitabine-based chemotherapy.
Reviewer 2 Report
Comment:
With 7 Figures, 4 Tables, and 215 references, Manuscript ID: pharmaceutics-1501275 [Type of manuscript: Review, Title: Smart Lipid-Based Nanosystems for Therapeutic Immune Induction against Cancers: Perspectives and Outlooks, by Authors: Seth-Frerich Fobian, Ziyun Cheng, Timo L. M. ten Hagen,] appeared comprehensive to envision the field. However, the authors should address the following 17 Specific comments and suggestions to improve organization and logic flow clarity.
Specific comments:
- The title: “Smart Lipid-Based Nanosystems for Therapeutic Immune Induction against Cancers: Perspectives and Outlooks” How did the authors define “smart?” In fact, neither smart nor novel came across the reader as the manuscript tracked down the past literature. Lines 899-906 touched but not in-depth.
- The abstract is written like an introduction. They should emphasize their outlooks, if any.
- Fig 1. The figure legend’s punctuations are confused.
- Figure 2. Venn diagram showing an overview of primary applications of available nanosystems used in immunotherapy. How did they obtain the quantifications of the patterns of the Venn diagram? If not quantifications, they should change the diagrams.
- Table 2: “Tight packing of lipids” – How did they define “tight?”
- Fig 4: “incorporated or conjugated agents” – Which are “conjugated agents and how?”
- Fig 6, Line 714 “directly target tumor transmembrane signaling pathways related to survival, (B)” – what specific did the authors mean? How? Nor was clear on liposomes “interact with the TME (including blood vessels).” What was the mechanism? Any specificity? The way of their illustration is confusing.
- Lines 720-730: not clear on how to target tumor cells. Anti-ephrin A2 scFvs?
- Lines 740-742: “docetaxel-loaded liposomes with ligands that bind specifically to vascular endothelial growth factor 2 receptors (VEGFR-2), a key angiogenic factor which is commonly upregulated in new peritumoral blood vessels, in breast cancer therapy [152].” This segment is typical across the version of the manuscript: no perspective or outlook from the authors.
- Table 3, line 851: A column showing efficacy is essential.
- Neither cytokine storm nor loss of target nor loss of control (dysfunctional) was fully addressed in the current version of the manuscript. Lines 899-906 touched but not in-depth. E.g., Lines 1023 – 1037: How long did the study measure? Another, e.g., lines 1093-1095, “Wang et al. reported that anchoring cyclized RGD peptide ligands upon PEGylated liposomes could elicit severe or lethal immune responses in a T cell-independent manner, due to repeated stimulation of the immune system [208].”
- Figure 7. Study design and description of in situ vaccination experiments using lipid nanoparticles. All of these procedures should be supported by relevant citations.
- Fig 7C & D: 10 days or 30 days?
- Lines 1060-1063: “Autologous total tumor mRNA and liposomal (DOTAP) pp65 full-length lysosomal-associated membrane protein (LAMP)-mRNA-loaded vaccines were designed to incorporate with surgery and radiation therapy.” How did they determine “total tumor mRNA?”
- Line 1087: A lack of discussions on “consider whether therapies show synergy according to the intrinsic mechanisms thereof” appears disappointing in the outlook, a way to illustrate “smart” that should be emphasized with the specific quantification of the synergy in a spatiotemporal model as the authors claimed on the title – which is the only way to make compelling arguments.
- Grammar errors. Not exhaustively listed here, but, with, E.g., “Even with these advances in preclinical liposomal nanoparticle immune therapy research, there are, at the time of writing, still none which have been approved by the FDA.” Lines 1096 – 1097.
- Lines 1132-1138: “clear optical imaging” has not been sufficiently addressed in the current version but is listed here in conclusion.
